# Investigating working memory updating processes of the human subcortex using 7T MRI

**Anne C Trutti[1,2]\*, Zsuzsika Sjoerds[2], Russell J Boag[1], Solenn LY Walstra[1], Steven Miletić[1,2], Scott JS Isherwood[1], Pierre-Louis Bazin[3], Bernhard Hommel[4], Sarah Habli[5], Desmond HY Tse[5], Asta K Håberg[5], Birte U Forstmann[1]**

[1]Integrative Model-Based Neuroscience Research Unit, University of Amsterdam, Amsterdam, Netherlands; [2]Cognitive Psychology Unit, Institute of Psychology and Leiden Institute for Brain and Cognition, Leiden University, Leiden, Netherlands; [3]Full brain picture Analytics, Leiden, Netherlands; [4]Department of Psychology, Shandong Normal University, Jinan, China; [5]Norwegian University of Science and Technology, Trondheim, Norway

**\*For correspondence:** annetrutti@gmail.com

**Competing interest:** The authors declare that no competing interests exist.

**Sent for Review** 08 April 2024
**Preprint posted** 11 April 2024
**Reviewed preprint posted** 29 July 2024
**Reviewed preprint revised** 22 January 2025
**Version of Record published** 25 June 2025

## eLife Assessment

This **valuable** study uses high-field fMRI to test the hypothesized involvement of subcortical structures, particularly the striatum, in updating working memory. The study overcomes limitations of prior work by applying high-field imaging with a more precise definition of regions of interest in the brain. Thus, the empirical observations are of use to specialists interested in working memory gating or the reference back task specifically. The evidence is generally **solid**, but strong conclusions on dopaminergic contributions must await additional work using molecular imaging or related techniques.

**Abstract** A growing body of research suggests that dopamine is involved in working memory updating and that the striatum takes up a critical role in the subprocess of working memory gating . In this study, we investigated subcortical – in particular, possible dopaminergic – involvement in working memory updating subprocesses using the reference-back task and ultrahigh field 7 Tesla functional magnetic resonance imaging (fMRI). Using a scanning protocol optimized for BOLD sensitivity in the subcortex, we found no evidence of subcortical activation during working memory gate opening, predominantly activations in frontoparietal network regions, which challenges the idea of a striatal gating mechanism. However, during gate closing, subcortical activation was observed. Furthermore, a ready-to-update mode demonstrated large-spread subcortical activation, including basal ganglia nuclei, suggesting that the basal ganglia are engaged in general updating processes rather than specifically controlling the working memory gate. Moreover, substituting new information into working memory elicited activation in dopamine-producing midbrain regions along with the striatum, thalamus, and prefrontal cortex, indicating engagement of the basal ganglia-thalamo-cortical loop possibly driven by (potential) dopaminergic activity. These findings expand our understanding of subcortical regions involved in working memory updating, shifting the focus from gate opening to substitution as a midbrain-driven updating process.

## Introduction

Dopamine exerts a profound neuromodulatory impact on neural activity and is capable of shaping behavior at a large scale. In particular, dopamine plays a central role in cognitive control, and leading theories detail how dopamine networks influence cognitive functions (*Cools and D'Esposito, 2011*; *Durstewitz and Seamans, 2008*; *Ott and Nieder, 2019*). These theories assign key roles to the meso-cortical and nigrostriatal dopaminergic pathways, which originate in the midbrain and project to the prefrontal cortex (PFC) and cingulate (for the mesocortical pathway), and the striatum (for the nigrostri-atal pathway). These pathways are believed to promote either cognitive stability (protection from distractors) or flexibility (ability to incorporate/adapt to new information), respectively (*Armbruster et al., 2012*; *Cools, 2006*; *Cools and D'Esposito, 2011*; *Durstewitz and Seamans, 2008*; *Goschke and Bolte, 2014*). This balance is known as the stability-flexibility trade-off (*Dreisbach, 2012*; *Dreis-bach and Fröber, 2019*; *Hommel, 2015*), influencing cognitive functions like working memory. For instance, maintaining stable working memory representations relies on PFC activity, the effectiveness of which is closely tied to cortical dopamine levels controlled by the mesocortical pathway (*Durste-witz and Seamans, 2008*). By contrast, the ability to flexibly update working memory representations is thought to be indirectly regulated by dopamine through the basal ganglia, implicating mesolimbic and nigrostriatal pathways (*Chatham and Badre, 2015*; *Braver and Cohen, 2000*). Several studies have provided initial support for working memory updating via the basal ganglia (*Chatham and Badre, 2015*; *Murty et al., 2011*; *Nir-Cohen et al., 2020*; *Nir-Cohen et al., 2023*). Even though the processes involved in allowing new information into working memory (i.e. updating) depend on dopamine (*Cools and D'Esposito, 2011*; *Goldman-Rakic, 1995*), the neural basis of working memory updating remains unclear.

General working memory functioning has been robustly associated with the frontoparietal network (FPN), which consists of the dorsolateral and medial PFC (dl/mPFC) and the posterior parietal cortex (PPC) (*Jacob and Nieder, 2014*; *Mehta et al., 2000*; *Nir-Cohen et al., 2020*; *Nir-Cohen et al., 2023*; *Vallentin et al., 2012*). Additionally, several studies have implicated the basal ganglia in working memory encoding and maintenance (*Bedwell et al., 2005*; *Chang et al., 2007*), updating (*Chatham and Badre, 2015*; *Murty et al., 2011*), and gating (*Nir-Cohen et al., 2020*). However, the exact neural processes involved in the updating of working memory, especially those involving the subcortical structures, are still not fully understood. Much of this uncertainty can be attributed to the relatively low signal-to-noise ratios of functional magnetic resonance imaging (fMRI) scans within the subcortex, as well as the inherent biophysical characteristics of subcortical nuclei, such as the presence of iron. These factors pose challenges when using conventional fMRI protocols to investigate the subcortex. In this study, we addressed this gap by conducting ultrahigh field 7 Tesla (T) fMRI to shed light on the subcortical structures thought to underlie working memory updating subprocesses.

Possible mechanisms behind selectively gating new information into working memory, i.e., updating, are captured in the *prefrontal-cortex basal-ganglia working memory* (PBWM) model (*Frank et al., 2001*; *Hazy et al., 2007*; *O'Reilly and Frank, 2006*). Central to the PBWM model is its prop-osition of a dynamic gating mechanism, where activation of the striatum acts as the pivotal initiator to open the gate to working memory – resulting in disinhibition of the thalamus through inhibition of the substantia nigra pars reticulata (SNr) – which in turn modulates the stability of working memory representations in PFC.

Initial support for the PBWM model has been found in studies employing the reference-back paradigm (*Rac-Lubashevsky and Kessler, 2016a*) in combination with EEG (*Rac-Lubashevsky and Kessler, 2018*), dopaminergic manipulations (*Jongkees, 2020*), and fMRI (*Nir-Cohen et al., 2020*). The reference-back task is designed to identify the behavioral signatures of several subprocesses involved in working memory updating. These include working memory gate opening and closing, substituting new items into working memory, and a contrast between updating and maintenance modes (indicating readiness to either update or shield the contents of working memory, respec-tively). These processes are derived by presenting a stream of reference and comparison stimuli, which require updating and maintenance, respectively. Gate opening represents the active process of switching from a maintenance mode to a mode that *facilitates* updating. Thus, opening the gate takes place when cortical cell assemblies are prepared for new information to enter working memory in trials that may call for updating. Similarly, gate closing occurs when information needs to be maintained, and, thus, the working memory gate should be closed and a maintenance mode activated in order to

avoid interference. Substitution represents the process of replacing old information with new information and thus represents the *actual act* of updating working memory. The updating mode reflects an open-gate state (successive reference trials) independent of whether updating of information is required. These subprocesses are often conflated in working memory tasks that tap into the PBWM model, such as the standard n-back task, task-switching paradigms, and the AX-CPT task (*Ecker et al., 2010*; *Kessler, 2017*; *Kessler, 2017*; *Lewis-Peacock et al., 2018*). In contrast, the reference-back task orthogonally differentiates these subprocesses – specifically isolating the crucial process of substituting a single stimulus in working memory from other facilitating subprocesses – providing a more precise mapping to brain regions, particularly in the deeper subcortical areas.

*Nir-Cohen et al., 2020*, found striatal activity associated with gate opening and substitution with fMRI at 3T. Moreover, they observed no evidence of subcortical involvement in the other aspects of working memory updating, like gate closing. This implies that distinct neural mechanisms control gate opening and gate closing. Nir-Cohen et al. suggested the involvement of the mesocortical pathway as a possible explanation for the lack of striatal activity during gate closing in line with the *dual-state theory* (*Durstewitz and Seamans, 2008*; *O'Reilly, 2006*), which postulates that mesocortical dopaminergic neurons of the ventral tegmental area (VTA) modify PFC neuronal firing to influence the stability of working memory representations. This implies the possibility of a direct dopaminergic mechanism originating in the midbrain governing the switch from updating mode to maintenance mode in gate closing. Also, in the PBWM model, dopamine may take a key role in the selective updating of working memory as the striatal pathways of the basal ganglia are abundant in dopamine receptors, enabling dopamine to affect basal ganglia processes critically. The neural origin of the striatal dopamine is found in the midbrain's dopaminergic cell populations, VTA, and SN. In fact, a revised version of the PBWM model for Parkinson's disease patients (*Moustafa et al., 2008*) accounts for the interaction of dopamine with working memory by including the SN pars compacta (SNc), emphasizing the role of dopamine in working memory updating.

Since dopaminergic functioning in the brain is almost exclusively based on the activity of dopaminergic cell assemblies in the midbrain, the VTA, and SN – specifically SNc, these areas warrant special attention in explorations of dopamine-related processes. Initial fMRI evidence suggests a link between working memory and midbrain activity. There were indications of the involvement of the VTA and SN in the updating of context-related information (*D'Ardenne et al., 2012*). Additionally, the combined processing of information removal and updating (*Murty et al., 2011*) also implicated these midbrain structures. Moreover, *Hazy et al., 2006*, provide their biologically based cognitive architecture onto the PBWM model and, in fact, suggest that dopaminergic signals arising from midbrain nuclei, specifically the VTA and SNc, modulate learning when striatal gating to mediate PFC activity should occur.

The indication that dopaminergic midbrain activity might play a pivotal role in updating working memory representations hints at a significant overlap in the brain mechanisms responsible for updating representations in working memory and updating learned reward-based values. The same neural circuits have been associated with updating of value-based decision-making and reinforcement learning (*Corlett et al., 2022*; *Jocham et al., 2011*; *O'Reilly, 2006*), in which the dopaminergic midbrain signal known as the reward prediction error (RPE) plays a key role (*Montague et al., 1996*; *Schultz, 2013*; *Schultz et al., 1997*). The RPE signal represents a value-updating signal that occurs when a received reward differs from what was expected according to reinforcement learning models (*Miletić et al., 2021*; *O'Doherty et al., 2017*; *Sutton and Barto, 2018*). The signal is encoded by phasic dopaminergic signals in the VTA and SNc (e.g. *Cooper et al., 2012*; *Montague et al., 1996*; *O'Doherty et al., 2017*; *Schultz et al., 1997*), targeting PFC representations via the striatum, following the same neural route that has been proposed to be involved in working memory updating (*Cohen et al., 2002*; *Hazy et al., 2007*; *Murty et al., 2011*; *Ponzi, 2008*).

In this study, we aimed to extend the work of *Nir-Cohen et al., 2020*, by exploring the neural correlates of working memory updating subprocesses, particularly focusing on the subcortex. To achieve this, we employed a scanning and analysis protocol to improve the signal quality from subcortical nuclei. This protocol included ultrahigh field 7T fMRI with a scanning protocol tailored to meet the requirements for imaging of the subcortex (*de Hollander et al., 2017*; *Miletić et al., 2020*) and individually parcellated masks of several subcortical nuclei (*Bazin et al., 2020*). While Nir-Cohen et al.'s results on striatal gate opening align with the hypothesized role of the striatum in working memory gating, they do not offer enough evidence to definitively rule out the subcortex's involvement in other

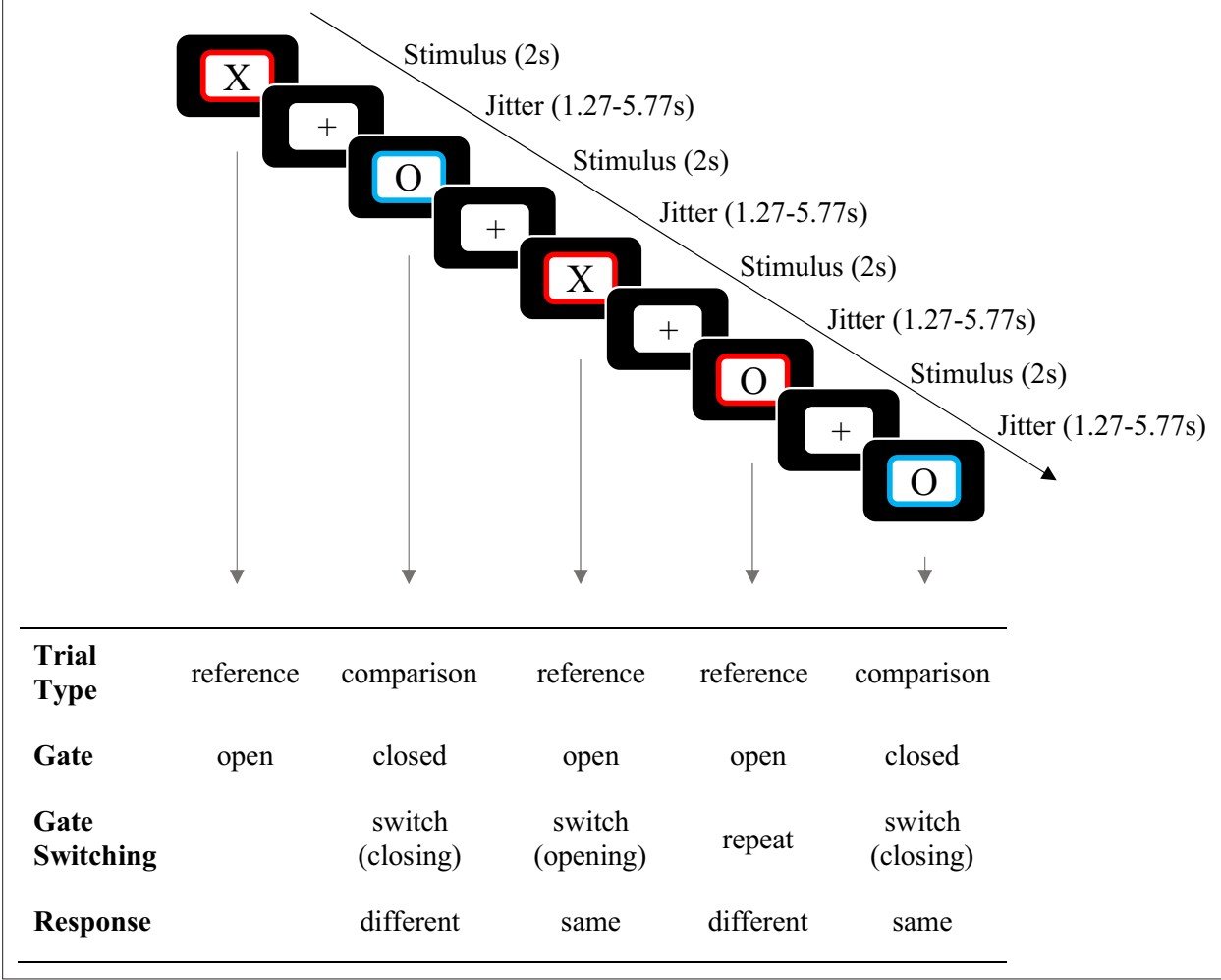

**Figure 1.** Example of the reference-back paradigm, including functional magnetic resonance imaging (fMRI) timing, as used in the experiment. In each trial, participants are tasked with indicating whether the probe stimulus ('X' or 'O') matches or differs from the stimulus presented in the most recent red frame, which serves as the working memory referent. In reference trials (red frame), participants are required to update their working memory with the currently displayed item. On the other hand, in comparison trials (blue frame), participants make the 'same/different' decision but do not update their working memory.

working memory update subprocesses. Hence, the present study specifically aimed to shed light on subcortical involvement in working memory subprocesses – associated with gating, substitution, and being in an updating mode – and to discern contributions from several subcortical structures. In addition to investigating subcortical nuclei associated with the basal ganglia-thalamo-cortical loop, we hypothesized that midbrain nuclei containing dopaminergic neurons (VTA and SN) might play a crucial role in several working memory updating subprocesses, with their activation in each contrast indicating different neural mechanisms. Precisely, in light of the findings from Nir-Cohen et al. discussed earlier, we tested whether the VTA demonstrates enhanced activity during gate closing as postulated by the dual-state theory (*Durstewitz and Seamans, 2008*). Furthermore, we aimed to investigate evidence of neural correlates of working memory substitution in the midbrain, specifically in the VTA and/or SN, which would support the idea that a phasic dopaminergic signal contributes to the act of changing working memory content, akin to the process of value updating following RPE. In contrast, VTA and/or SN activity in the gate opening process would lend support to theoretical accounts suggesting that a phasic dopaminergic signal opens the gate to working memory and, this way, facilitates (potential) subsequent working memory updating. Along these lines, activation in the VTA and/or SN during an updating mode may support the idea that dopaminergic signals are actively engaged in keeping the gates to working memory open.

**Table 1.** Hierarchical descriptive statistics for the behavioral data.

| | | | Response time | | Accuracy | |
|---|---|---|---|---|---|---|
| Trial type | Switch type | Response type | *Mean (s)* | *SD* | *Mean (%)* | *SD* |
| Reference | Repeat | Same | 0.751 | 0.243 | 97.205 | 0.164 |
| | | Different | 0.957 | 0.269 | 95.298 | 0.211 |
| | Switch | Same | 0.807 | 0.274 | 94.560 | 0.227 |
| | | Different | 1.012 | 0.291 | 91.001 | 0.286 |
| Comparison | Repeat | Same | 0.776 | 0.262 | 93.085 | 0.254 |
| | | Different | 0.861 | 0.253 | 93.693 | 0.243 |
| | Switch | Same | 0.76 | 0.216 | 96.196 | 0.191 |
| | | Different | 0.896 | 0.255 | 96.357 | 0.187 |

Taken together, this 7T fMRI study was designed to shed light on subcortical – in particular dopaminergic – contributions to working memory updating subprocesses in the human brain (***Figure 1***).

## Results

### Behavioral results

The results of the behavioral analyses of the reference-back task are presented in the appendix (***Appendix 1—tables 3–5***). Descriptive statistics are reported in ***Table 1***.

***Table 2*** illustrates the group-averaged mean response time (RT) and accuracy for each condition of the experimental design. The overall mean RT was 0.86 s (mean RT for correct responses was 0.85 s), and the overall accuracy on the task was 94.7%, indicating that participants understood the task instructions. Correct responses were, on average, faster than errors (Δ=0.169 s, t=12.8, df = 149, p<0.001, d=0.61).

For mean RT, there were significant main effects of trial type, switch type, and response. Additionally, the two-way interactions but not the three-way interactions were significant. For accuracy, there were no significant main effects. However, we found two-way interactions between trial type and switch type paired with trial type and response. There was no three-way interaction (***Appendix 1— Tables 3–5***, ***Appendix 1—figure 2***).

Responses were slower and less accurate on reference/switch (RT: M=0.916 s, SEM = 0.017; accuracy: M=0.920, SEM = 0.008) than reference/repeat trials (RT: M=0.856 s, SEM = 0.016; accuracy: M=0.962, SEM = 0.005), which represents the behavioral costs of working memory gate opening.

Responses were not substantially slower on comparison/switch (M=0.829 s, SEM = 0.012) than comparison/repeat trials (M=0.821 s, SEM = 0.008), suggesting no behavioral RT cost of closing the working memory gate. Interestingly, responses were more accurate on comparison/switch trials (M=0.962, SEM = 0.005) compared to comparison/repeat trials (M=0.933, SEM = 0.008), indicating

**Table 2.** Contrast weights for defining the four distinct working memory updating subprocessess.

| Trial type | reference | | | | comparison | | | |
|---|---|---|---|---|---|---|---|---|
| Gate switch | repeat | | switch | | repeat | | switch | |
| Response | same | different | same | different | same | different | same | different |
| **Contrast** | | | | | | | | |
| Gate opening | - | - | + | + | | | | |
| Gate closing | | | | | - | - | + | + |
| Substitution | - | + | | | + | - | | |
| Updating mode | + | + | | | - | - | | |

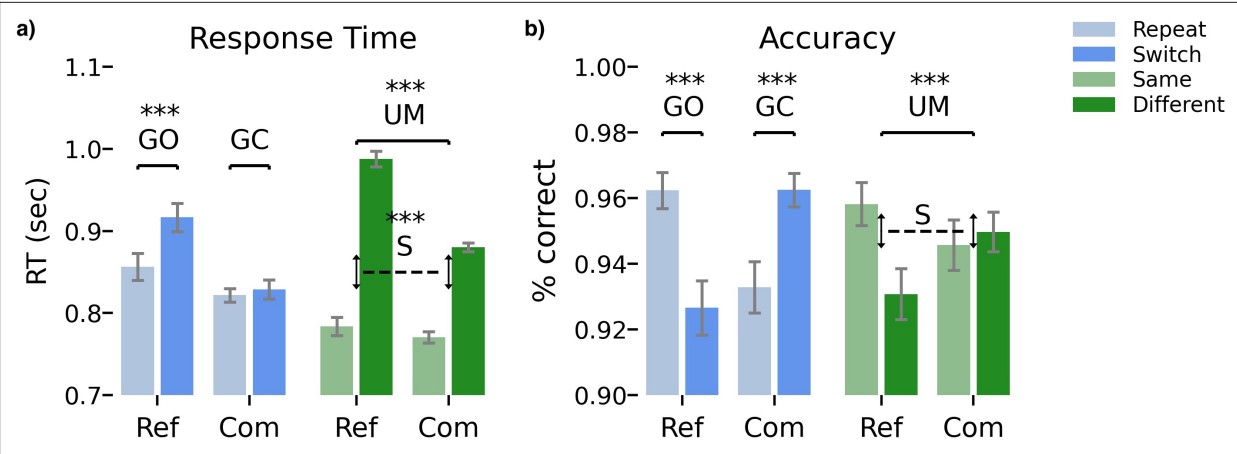

**Figure 2.** The figure illustrates the mean response times (and standard error of the mean) in relation to two factors. (**a**) whether the condition was switched or repeated and (**b**) whether the stimulus/response matched the previous reference stimulus or differed from it. The figure also displays the associated behavioral contrasts and their significance levels. See Reference-back task for detailed information on how the contrasts were computed. Ref = reference trials; Com = comparison trials; GO = gate opening; GC = gate closing; S=substitution; UM = updating mode. ***=p<0.001.

The online version of this article includes the following source data for figure 2:

**Source data 1.** Behavioral measures from the experimental design.

that gate closing did not produce any behavioral cost, in fact an accuracy gain in keeping the gate closed.

For the substitution contrast, the difference between 'different' and 'same' responses on repeated reference trials was slower and less accurate (RT: Δ=0.207 s, SEM = 0.017; accuracy: Δ=–0.019, SEM = 0.010) than the difference between 'different' and 'same' responses in repeated comparison trials (RT: Δ=0.082 s; SEM = 0.010; accuracy: Δ=0.007, SEM = 0.012). This suggests costs of both RT and accuracy in working memory substitution processing.

Finally, responses were slower but more accurate on repeated reference trials (RT: M=0.856 s, SEM = 0.016; accuracy: M=0.962, SEM = 0.005) responses compared with repeated comparison trials (RT: M=0.821 s, SEM = 0.008; accuracy: M=0.933, SEM = 0.008), reflecting the RT cost of being in a general updating mode.

## fMRI results

### Whole-brain GLMs

Whole-brain analyses were conducted to explore the brain activations associated with gate opening, gate closing, substitution, and updating (**Figure 2**).

### Gate opening

Gate opening was associated with large bilateral clusters of activation across the cortex. Specifically, we found increased activation in frontal cortical regions, including the mPFC and dlPFC, as well as the somatosensory and motor cortices (**Figure 3**; **Table 3**). Gate opening was also associated with increased activation in posterior parietal regions, including the precuneus cortex and left parietal lobe. The occipital lobe, including the left and right fusiform cortex, also showed increased activation during gate opening. Subcortically, gate opening was associated with clusters of increased activation in the thalamus, in particular a large cluster in the left thalamus. These findings are consistent with **Nir-Cohen et al., 2020**. Additionally, we found a large cluster of activation in the brainstem and midbrain regions (see Figure 7), covering areas such as the left and right red nucleus, periaqueductal gray, superior peduncle, and left subthalamic nucleus (STN).

### Gate closing

Gate closing was associated with a few clusters of cortical activations, in line with **Nir-Cohen et al., 2020**. Our largest cluster covered much of the left PPC; another cluster occurred in the left dlPFC. In contrast to findings by Nir-Cohen et al., there was no activation in the right hemisphere except for

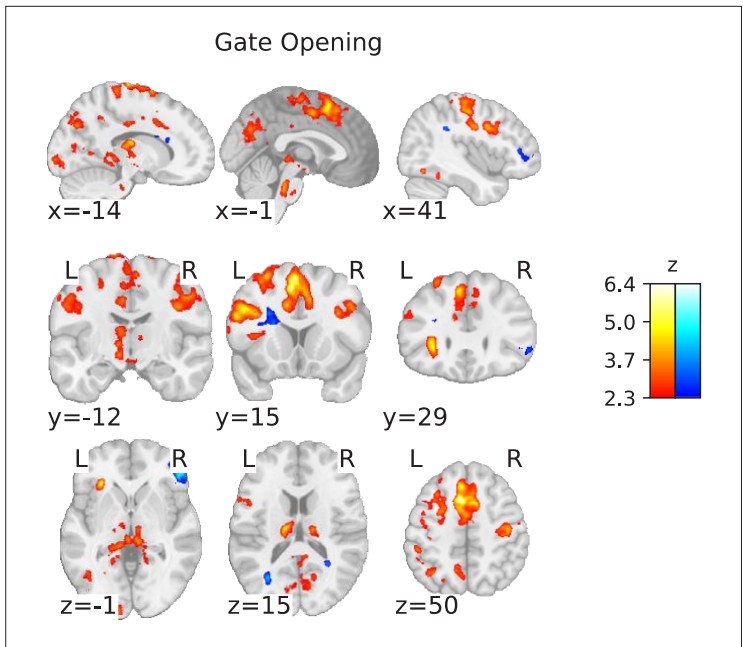

**Figure 3.** Statistical parametric map of the 'gate opening' contrast with a threshold determined using the family-wise error rate (FWER) method (p<0.05; corresponding to a threshold of z=2.3).

relatively small activation clusters in the bilateral occipital fusiform gyrus. In addition, gate closing was associated with increased activity in the left inferior temporal gyrus and cerebellum (*Figure 4*; *Table 3*).

## Substitution

Substitution was associated with several large clusters of activation. There was increased activation in the premotor cortex and a substantial portion of the PFC, including Brodman area (BA) 8 (dlPFC), BA 24, and BA 32 (mPFC). There was increased activation in the pre-supplementary motor area (preSMA), the superior and inferior frontal gyrus, the left and right PPC, and the left inferior parietal lobule (IPL). The subcortex showed heightened activation in both the right and left striatal regions, including the caudate and putamen. A complete list of all active clusters is given in *Table 3*. The pattern of activation associated with substitution is broadly consistent with *Nir-Cohen et al., 2020*. However, we found a considerably greater extent of cortical activation, along with subcortical activation not observed by *Nir-Cohen et al., 2020* (*Figure 5* and Figure 7).

## Updating mode

The updating mode was associated with increased activation in frontal and posterior parietal regions. The frontal activation covered the left and right medial to dorsolateral prefrontal regions (from the preSMA, over the superior and middle frontal gyrus, to the inferior frontal gyrus). The posterior parietal cluster covered the IPL and stretched from the supramarginal gyrus, medially toward the precuneus cortex, and ventrally along the angular gyrus toward the medial temporal gyrus. There was increased activation in occipital regions, including the fusiform gyrus and the intra- and supracalcarine cortex. In the subcortex, clusters of increased activation were found in the left and right putamen, caudate nuclei, and the right thalamus. This pattern of activation is consistent with *Nir-Cohen et al., 2020*, but extends their findings to a broader range of cortical (*Figure 6*; *Table 3*) and subcortical regions (*Figure 7*).

## ROI analyses

We conducted three sets of region-of-interest (ROI) analyses, as described in Regions-of-interest. The main ROI analyses included ROI-wise GLMs based on the individually parcellated masks derived from MASSP (*Bazin et al., 2020*). In addition, we conducted an additional ROI analysis into subdivisions

**Table 3.** List of peak activation in MNI coordinates from the whole-brain analysis.

| | | hem | Voxels | MNI x | y | z | Z |
|---|---|---|---|---|---|---|---|
| *Gate opening* | | | | | | | |
| | preSMA | l | 13380 | –6.0 | 12.0 | 50.0 | 6.387 |
| | Occipital fusiform gyrus | l | 8915 | –29.0 | –63.0 | –7.0 | 5.144 |
| | MFG | r | 849 | 34.0 | 9.0 | 29.0 | 5.129 |
| | Insular | l | 724 | –30.0 | 29.0 | –1.0 | 5.025 |
| | Precuneus cortex | l | 5482 | 7.0 | –67.0 | 25.0 | 4.422 |
| | Brainstem | | 841 | 0.0 | –33.0 | –29.0 | 4.338 |
| | Primary somatosensory cortex | r | 2043 | 41.0 | –16.0 | 50.0 | 4.219 |
| | M1 | l | 976 | –27.0 | –28.0 | 64.0 | 4.064 |
| | Primary somatosensory cortex | l | 566 | –53.0 | –14.0 | 40.0 | 3.958 |
| | M1 | r | 81 | 11.0 | –24.0 | 81.0 | 3.363 |
| | Primary somatosensory cortex | r | 34 | 66.0 | –8.0 | 30.0 | 3.284 |
| | Primary motor cortex | l | 92 | –35.0 | –20.0 | 45.0 | 3.188 |
| *Gate closing* | | | | | | | |
| | Inferior parietal cortex | l | 3769 | –27.0 | –64.0 | 36.0 | 4.560 |
| | Inferior temporal gyrus | l | 716 | –52.0 | –65.0 | –12.0 | 4.376 |
| | VI (cerebellum) | r | 2418 | 28.0 | –61.0 | –28.0 | 4.316 |
| | Fusiform gyrus | l | 957 | –33.0 | –80.0 | –20.0 | 4.284 |
| | V (cerebellum) | l | 1680 | –50.0 | 42.0 | 9.0 | 4.091 |
| | Fusiform gyrus | r | 14 | 35.0 | –79.0 | –21.0 | 3.070 |
| *Substitution* | | | | | | | |
| | M1 | l | 68499 | –45.0 | –13.0 | 38.0 | 6.834 |
| | Putamen | l | 1813 | –20.0 | 5.0 | 0.0 | 5.530 |
| | BA 9 (dlPFC) | l | 714 | –22.0 | 56.0 | 33.0 | 4.763 |
| | Insular | l | 108 | –31.0 | 17.0 | 10.0 | 3.389 |
| | Brainstem | | 32 | –19.0 | –32.0 | –33.0 | 3.184 |
| | M1 | r | 15 | 25.0 | –25.0 | 68.0 | 2.974 |
| | BA 9 (dlPFC) | l | 21 | –31.0 | 41.0 | 45.0 | 2.892 |
| | Parahippocampal gyrus | l | 12 | –28.0 | –30.0 | –20.0 | 2.825 |
| | Visual cortex V1 | r | 12 | 29.0 | –62.0 | 5.0 | 2.803 |
| | Cingulate gyrus | r | 15 | 5.0 | –51.0 | 19.0 | 2.728 |
| | Inferior parietal lobule | l | 13 | –57.0 | –44.0 | 34.0 | 2.652 |
| | Visual cortex V4 | r | 19 | 22.0 | –71.0 | –1.0 | 2.633 |
| *Updating mode* | | | | | | | |
| | BA 40 (PPC) | l | 24635 | –56.0 | –41.0 | 55.0 | 8.084 |
| | MFG | l | 19609 | –36.0 | 0.0 | 65.0 | 6.160 |
| | BA 9 (dlPFC) | l | 5254 | –44.0 | 38.0 | 33.0 | 5.816 |
| | Fusiform gyrus | r | 12503 | 30.0 | –70.0 | –17.0 | 5.776 |

*Table 3 continued on next page*

*Table 3 continued*

| | | | MNI | | | |
|---|---|---|---|---|---|---|
| Putamen | l | 1486 | −21.0 | 15.0 | −2.0 | 4.983 |
| Caudate | r | 1702 | 17.0 | 13.0 | 16.0 | 4.384 |
| Inferior temporal gyrus | r | 45 | 49.0 | −38.0 | −16.0 | 3.162 |
| Precuneus cortex | r | 19 | 18.0 | −60.0 | 31.0 | 2.754 |

of the dopaminergic midbrain regions through the exploration of masks derived from a probabilistic atlas. A supplementary exploration into clusters of activation within ROIs based on the whole-brain GLMs was also performed for comparison with *Nir-Cohen et al., 2020* (see A1.1). Hence, the first ROI-wise GLMs ('ROI-wise GLMs on individually parcellated masks') emphasized individual delineations and thus provided high precision with regard to individual anatomy. The post hoc ROI-wise GLMs provided information on the possible differential involvement of subnuclei of the VTA and substantia nigra (SN), however, at the cost of individual anatomical detail and will be referred to as 'ROI-wise GLMs based on VTA/SN subdivisions atlas' in the following. Results of the ROI analyses are illustrated in *Figures 8 and 9*, *Tables 2 and 3*.

## ROI-wise GLMs on individually parcellated masks

ROI-wise GLMs analyses demonstrated that subcortical regions play a role in each experimental contrast. The main findings for each experimental contrast are summarized below and depicted in *Figure 8* and *Table 4*.

## Gate opening

During gate opening, subcortical activation was limited to the right thalamus. Bayesian analyses provided only weak evidence for activity in the right thalamus, contrasting the observation of extensive bilateral thalamus activation in the whole-brain contrast. Notably, we found moderate evidence against activity in any other ROI in the basal ganglia and midbrain, along with weak evidence against activity in the right thalamus, right GPe, and right GPi.

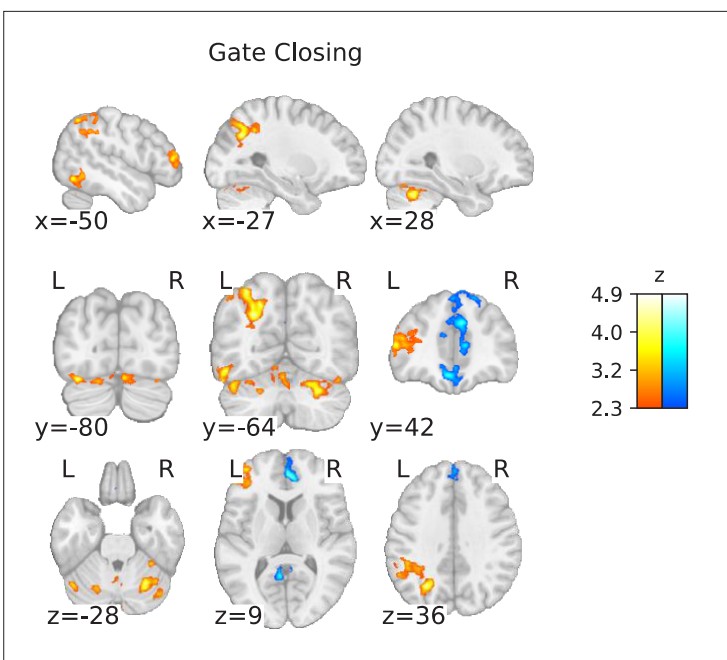

**Figure 4.** Statistical parametric map of the 'gate closing' contrast with a threshold determined using the family-wise error rate (FWER) method (p<0.05; corresponding to a threshold of z=2.3).

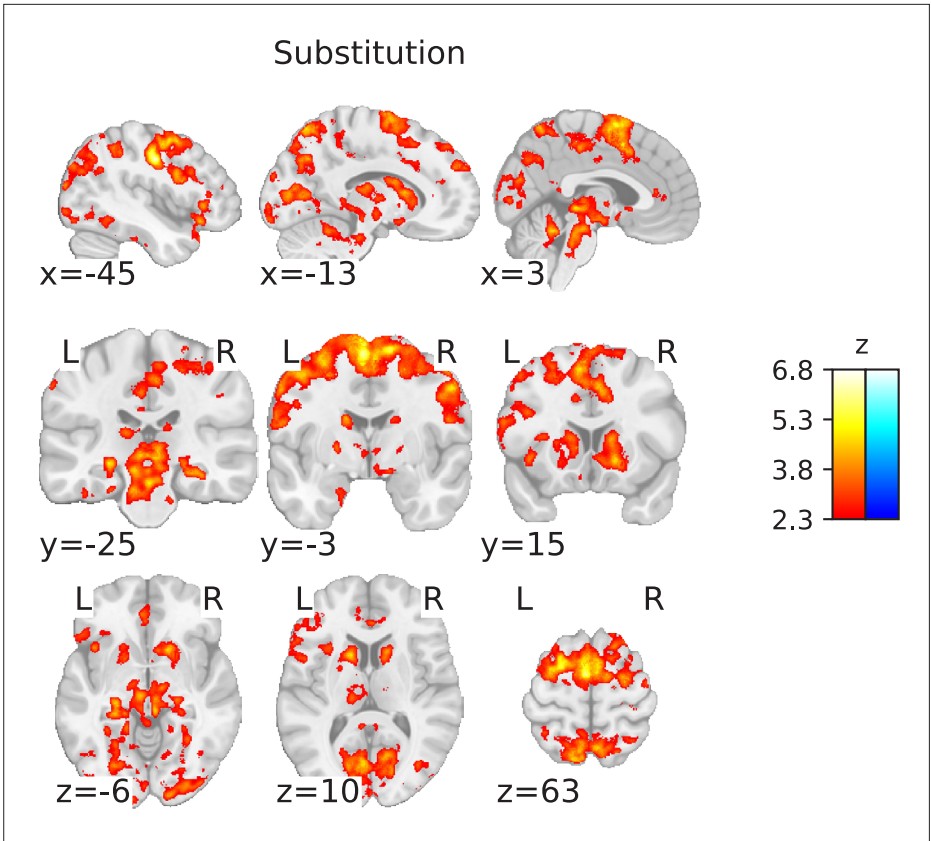

**Figure 5.** Statistical parametric map of the 'substitution' contrast with a threshold determined using the family-wise error rate (FWER) method (p<0.05; corresponding to a threshold of z=2.3).

### Gate closing

ROI-wise GLM results indicated moderate evidence for activity in the right GPe and weak evidence for activity in the right basal ganglia nuclei, namely right striatum and right STN, during gate closing. Moreover, bilateral midbrain activation of the SN was observed, indicated by weak evidence.

### Substitution

During substitution, the results of ROI-wise GLMs confirmed the engagement of subcortical structures, although there were variations in their individual contributions. Moderate evidence of activity was observed in the striatum bilaterally, in the right STN, and the right VTA. Moreover, weak evidence indicated activity in the thalamus, left GPe, left GPi, and left STN. Overall, Bayesian analyses did not indicate strong evidence against activity in any subcortical ROI during the process of substitution.

### Updating mode

Being in an updating mode was associated with both basal ganglia and midbrain activity. Precisely, ROI-wise GLM results provided Bayes factors suggesting strong evidence for activity in the left GPe and right STN. Yet, only moderate evidence for the right GPe, left GPi, and right SN. Furthermore, weak evidence from the ROI GLMs indicated activity of the thalamus, striatum, left STN, and right VTA. Overall, there were no Bayes factors indicating strong evidence against activity related to the updating mode in any ROI.

### ROI-wise GLMs based on VTA/SN subdivisions atlas

ROI-wise GLM analysis demonstrated that midbrain nuclei play a role in gate closing, substitution, and updating mode. The main findings for each experimental contrast are summarized below and depicted in *Figure 9* and *Table 5*.

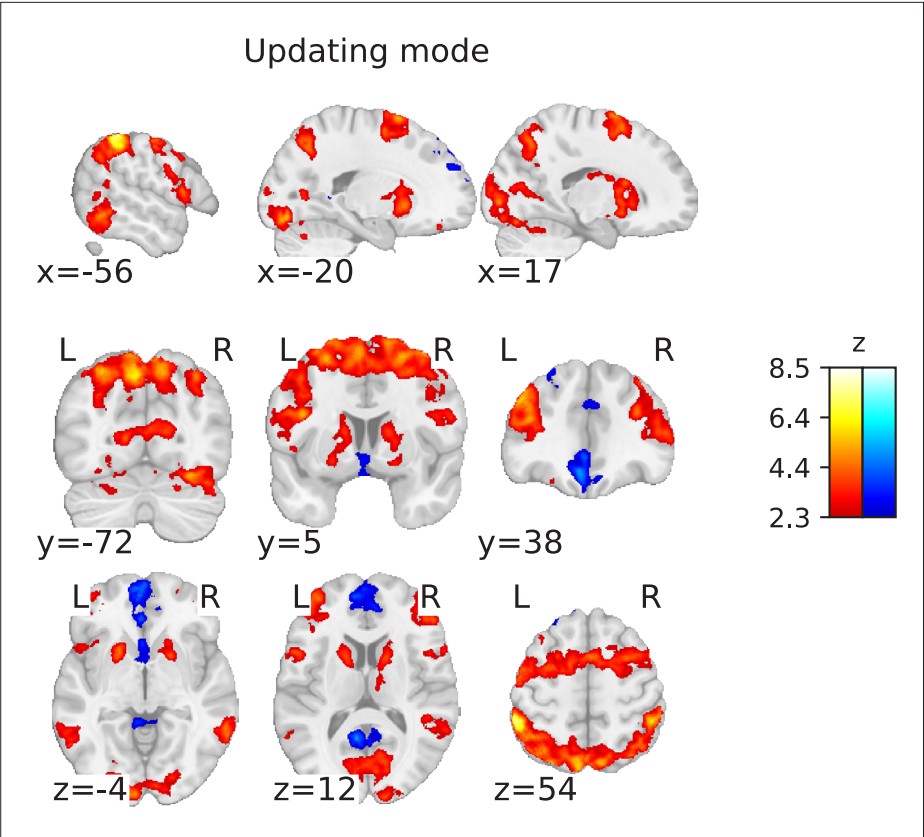

**Figure 6.** Statistical parametric map of the 'updating mode' contrast with a threshold determined using the family-wise error rate (FWER) method (p<0.05; corresponding to a threshold of z=2.3).

### Gate opening

Confirming results from the preceding ROI-wise GLM analysis, ROI-wise GLMs using the probabilistic atlas masks implicated no evidence for activity in midbrain dopamine-producing nuclei in gate opening, supported by moderate evidence against activity in any mask from *Pauli et al., 2018*.

### Gate closing

ROI-wise GLMs provided evidence for SNc activation during gate closing. There was strong evidence for activity in the right SNc and moderate evidence for activity in the left SNc. This high Bayes factor in favor of the right SNc suggests a dopaminergic involvement during gate closing. Furthermore, the results of the Bayesian analyses did not provide strong evidence against activity in any ROI during gate-closing trials.

### Substitution

In line with the results from the ROI analyses based on MASSP delineations (see above), results implicated activity in the right VTA. Specifically, the right parabrachial pigmented (PBP) mask by *Pauli et al., 2018*, indicated activity with moderate evidence, suggesting an involvement of the dorsolateral nucleus of the VTA during substitution. In addition, BOLD signal change in the right SNc demonstrated a high Bayes factor. Both findings indicate dopaminergic involvement in the substitution process.

### Updating mode

Midbrain masks by *Pauli et al., 2018*, demonstrated an extremely high Bayes factor for the right SNc – insinuating very strong evidence for increased activity – moderate evidence for activation in the left SNc, along with weak evidence for activity in the right PBP during updating mode. Again, there were no Bayes factors that indicated strong evidence against activity related to the updating mode in any ROI.

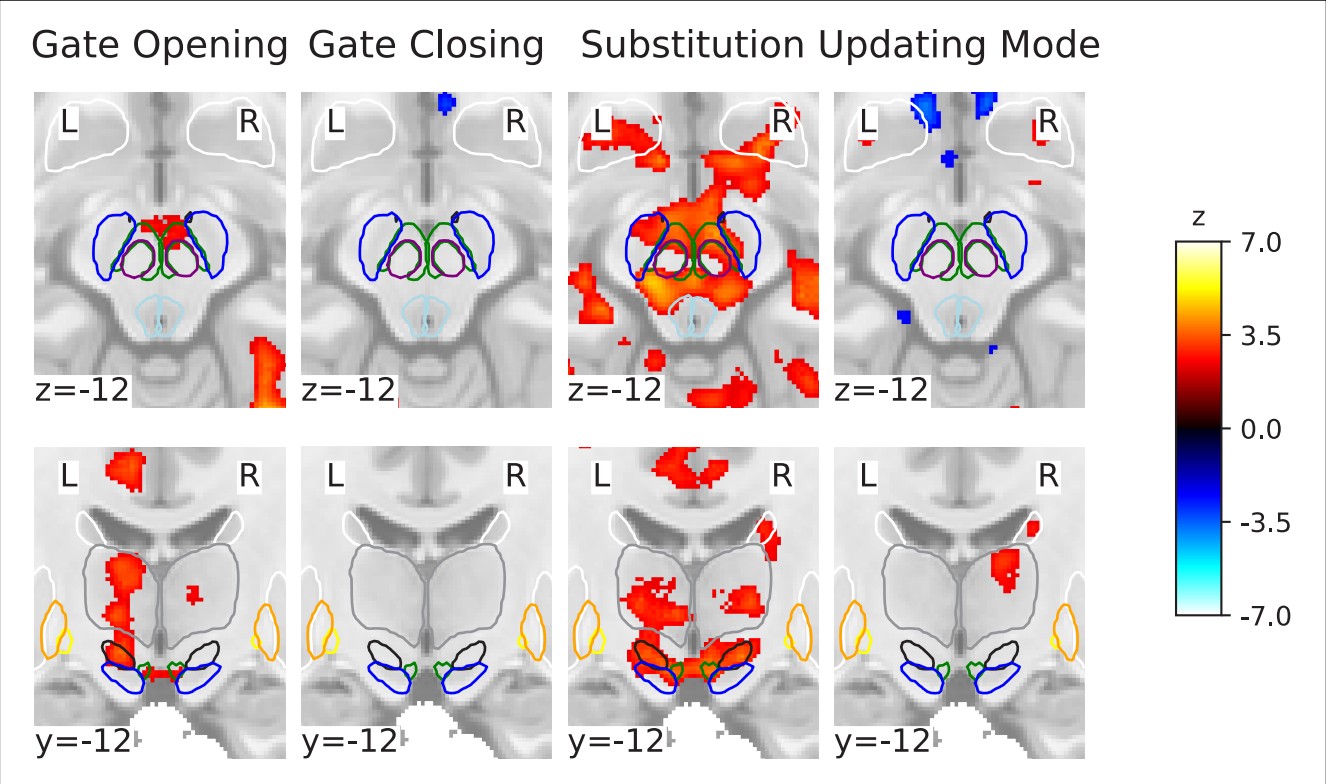

**Figure 7.** Subcortical statistical parametric mapping of the four contrasts with a focus on the midbrain (top row), and midbrain and thalamic regions (bottom row) using a family-wise error rate (FWER) threshold of 2.3. Coordinates are in MNI2009cAsym (1 mm) space. For orientation purposes, the MASSP atlas is overlaid: substantia nigra (SN) (blue), ventral tegmental area (VTA) (black), red nucleus (purple), subthalamic nucleus (green), periaqueductal gray (light blue), globus pallidus interna (GPi) (yellow), globus pallidus externa (GPe) (orange), striatum (white), and thalamus (gray).

In summary, no subcortical engagement except for the thalamus was revealed in the gate opening contrast. However, ROI-wise GLMs indicated only weak evidence for the right thalamus during gate opening, rendering our results on gate-opening-related thalamic activity inconclusive. Notably, we observed moderate evidence against the involvement of most ROIs in the basal ganglia and midbrain during gate opening. Nevertheless, during gate closing, substitution, and updating mode, evidence did suggest activity in the basal ganglia and midbrain. It appears, however, that basal ganglia nuclei are differently engaged across gate closing, substitution, and updating mode contrasts. Reasonable evidence for striatal engagement was only found during working memory substitution trials. In the pallidum, gate closing-associated activity implicated the right GPe, and there was very strong evidence for the left GPe in updating mode. Additionally, results implicated the involvement of the right STN in updating mode and in substitution.

Also, the regions of the midbrain appear to be differently engaged across the three contrasts. The right SN was active in the updating mode while there was only weak evidence for increased SN activation during gate closing and none during substitution. However, using the probabilistic mask by *Pauli et al., 2018*, data indicated strong evidence for SNc activation during updating mode and strong evidence for SNc engagement during gate closing and substitution. Interestingly, evidence for engagement of the left SNc was substantially lower in all three contrasts. Furthermore, results implicated the right VTA mask by MASSP (*Bazin et al., 2020*) in substitution. Interestingly, considering the analyses using the probabilistic masks, this activation appears to be driven by the (right) PBP, a VTA component nucleus.

## Discussion

The present study aimed to shed light on the neural substrates of working memory subprocesses, particularly focusing on the subcortex. As an extension of the work by *Nir-Cohen et al., 2020*, we

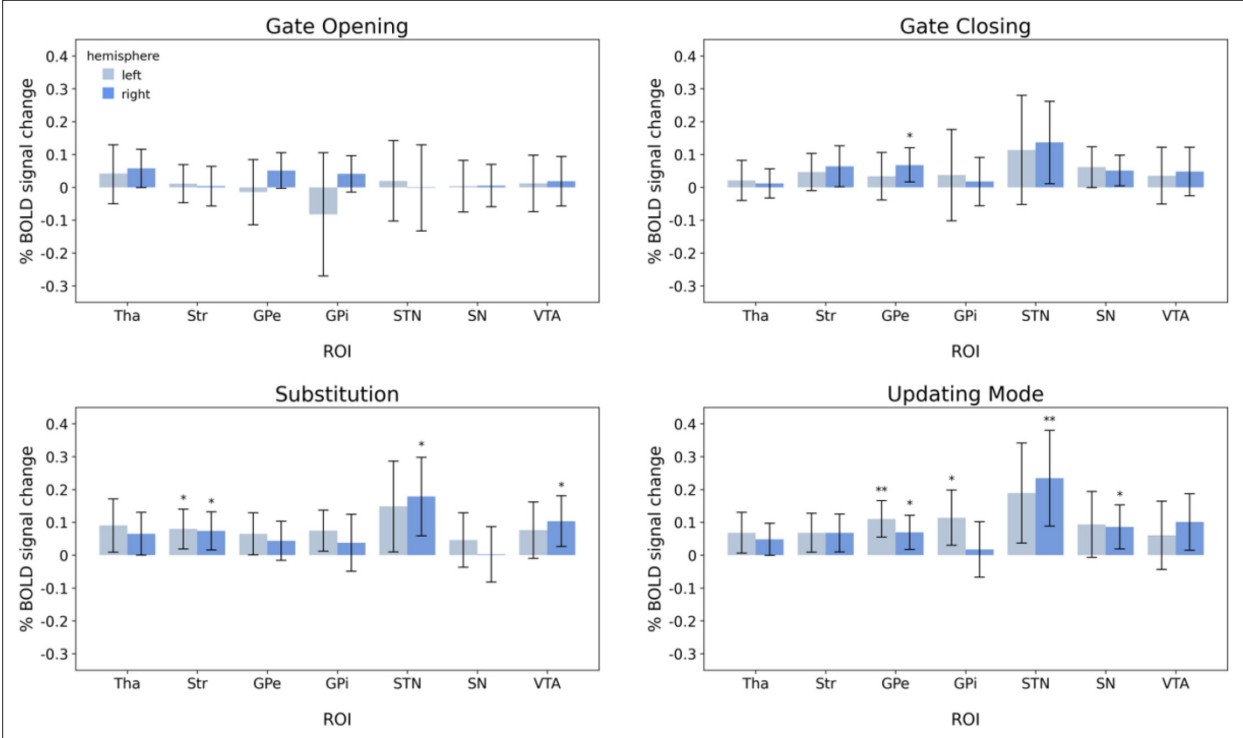

**Figure 8.** Results of the region-of-interest-wise GLMs using the individually parcellated masks derived from MASSP. Error bars represent the 95% credible intervals. Abbreviations indicate thalamus (Tha), striatum (Str), globus pallidus externa (GPe), globus pallidus interna (GPi), subthalamic nucleus (STN), substantia nigra (SN), and ventral tegmental area (VTA). * indicate moderate evidence and ** indicate strong evidence that the observed BOLD signal change is larger than 0.

employed the reference-back paradigm in conjunction with 7T fMRI, including a scanning and analysis protocol optimized for the subcortex, to precisely discern contributions from several subcortical structures to working memory updating subprocesses associated with gating, substitution, and being in an updating mode. In addition to investigating nuclei in the basal ganglia-thalamo-cortical loop, we hypothesized that midbrain nuclei containing dopaminergic neurons, the VTA, and SN might play a pivotal role in working memory updating subprocesses.

The whole-brain analysis not only revealed a substantially broader range of brain activation in both the cortex and subcortex but also provided more details compared to previous work (***Nir-Cohen et al., 2020***). More precisely, consistent with the work by ***Nir-Cohen et al., 2020***, we found that substitution and being in a general mode of updating show increased activation in all regions belonging to the FPN. Overall, the components of the FPN seem to be involved across all contrasts. However, our data suggests that each working memory subprocess differently engages the individual FPN components: posterior parietal regions play a greater role in gate-switching, while substitution and updating primarily recruit the frontal regions and subcortex. This observation implies that gate-switching may primarily serve as a selective attention process, a view supported by ***van Schouwenburg et al., 2014***, through their findings on attention-modulated cortical neural activities involved in gate-switching mechanisms.

Furthermore, our whole-brain analysis identified thalamic activation in the process of opening the gate to working memory, working memory substitution, and the updating mode – aligning with Nir-Cohen et al.'s findings. Nonetheless, they found thalamic activity only in an ROI analysis and not their whole-brain analysis, demonstrating our enhanced signal and spatial resolution. Additionally, our whole-brain analysis revealed activation in distinct subcortical regions: the brainstem during gate opening, cerebellar regions during gate closing, and midbrain during gate opening, substitution, and updating mode. Notably, during substitution and updating mode, there was activation in striatal subregions – specifically the caudate nucleus and putamen – particularly extensive bilateral striatal activation during substitution, aligning with the PBWM model's actual updating process.

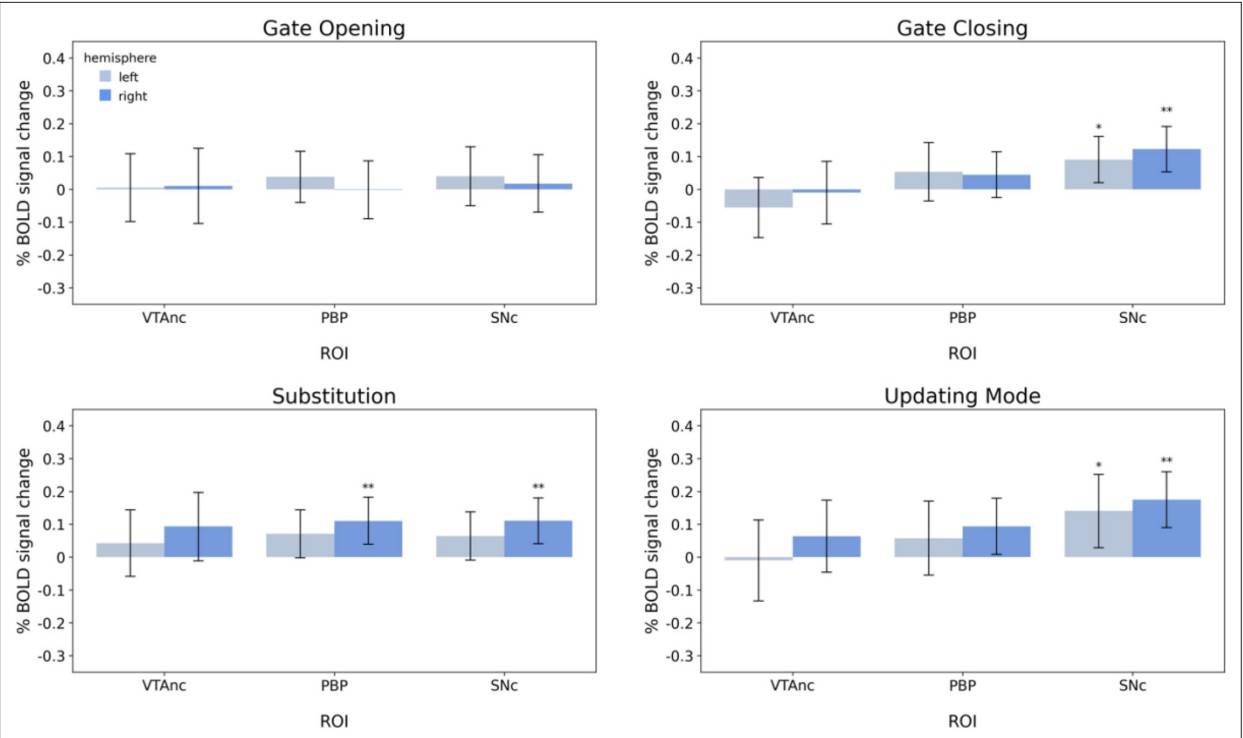

**Figure 9.** Results of the region-of-interest-wise GLMs using the probabilistic atlas from *Pauli et al., 2018*. Error bars represent the 95% credible intervals. Abbreviations indicate ventral tegmental area (VTA) nucleus, parabrachial pigmented (PBP) nucleus, and substantia nigra pars compacta (SNc). * indicate moderate evidence and ** indicate strong evidence that the observed BOLD signal change is larger than 0.

We extend previous work by investigating individual masks of subcortical ROIs using the automated subcortical parcellation algorithm MASSP (*Bazin et al., 2020*), including regions covering the dopaminergic midbrain – additionally supported by ROI analyses using masks from *Pauli et al., 2018* – each considered bilaterally. Together, they revealed activation in the basal ganglia and midbrain during gate closing, substitution, and updating mode, albeit with differing degrees of engagement across these contrasts. Notably, no subcortical engagement was observed in gate opening, except for weak evidence for the right thalamus. This still might indicate a possible involvement of the thalamus during gate opening since the whole-brain analysis yielded large bilateral activation clusters in the thalamus. Furthermore, while the Bayesian analysis provided moderate evidence against any subcortical ROI in gate opening, the evidence against the right thalamus was nihil. All these factors render the results for thalamic involvement during gate opening inconclusive. However, it should be noted that activation limited to a thalamic subnucleus may explain the observed findings from the ROI analysis. Precisely, it is possible that an existing activation was negated as the BOLD signal change was averaged across all voxels included in a specific mask. Our data, therefore, did not definitively support or contradict thalamic involvement in gate opening, underscoring the necessity for individually delineated masks for subnuclei in the thalamus in future studies.

Our findings refine prominent neural theories of working memory gating by showing that, rather than controlling gating, the basal ganglia's role may be more specific to the actual act of updating working memory representations with new information and a sustained open-gate state. The reasons for this are twofold. First, there was an absence of evidence for striatal participation in gate opening. Second, we found midbrain and basal ganglia activation during substitution, suggesting a neural signature of midbrain activation – potentially dopaminergic – of the basal ganglia-thalamo-cortical loop. These findings significantly advance our understanding of the cortical and subcortical neural basis of working memory updating subprocesses.

## No support for striatal gate opening

The first aspect of the twofold revelation, indicating a role for the basal ganglia in working memory updating that differs from what was originally postulated, is the absence of evidence for striatal gate

**Table 4.** Results from the Bayesian one-sample t-test on the beta values derived from the region-of-interest (ROI)-wise GLMs are reported for each contrast and ROI.

Moderate or higher evidence for ROI activity is indicated by bold font. Bayes factors favoring the alternative hypothesis are reported under $BF_{10}$, and Bayes factors favoring the null hypothesis in column $BF_{01}$.

| | ROI | hem | $BF_{10}$ | $BF_{01}$ | Error % |
|---|---|---|---|---|---|
| *Gate opening* | Tha | l | 0.287 | 3.489 | 0.037 |
| | | r | 1.070 | 0.934 | 0.023 |
| | Str | l | 0.197 | 5.072 | 0.041 |
| | | r | 0.185 | 5.407 | 0.041 |
| | GPe | l | 0.192 | 5.213 | 0.041 |
| | | r | 0.952 | 1.050 | 0.024 |
| | GPi | l | 0.266 | 3.755 | 0.038 |
| | | r | 0.510 | 1.962 | 0.031 |
| | STN | l | 0.193 | 5.184 | 0.041 |
| | | r | 0.184 | 5.440 | 0.041 |
| | SN | l | 0.184 | 5.421 | 0.041 |
| | | r | 0.186 | 5.372 | 0.041 |
| | VTA | l | 0.190 | 5.250 | 0.041 |
| | | r | 0.207 | 4.831 | 0.040 |
| | Tha | l | 0.229 | 4.364 | 0.039 |
| | | r | 0.208 | 4.799 | 0.040 |
| | Str | l | 0.643 | 1.556 | 0.028 |
| | | r | 1.239 | 0.807 | 0.021 |
| | GPe | l | 0.275 | 3.636 | 0.038 |
| | | r | **3.582** | **0.279** | $6.359×10^{-7}$ |
| | GPi | l | 0.211 | 4.746 | 0.040 |
| | | r | 0.205 | 4.874 | 0.040 |
| | STN | l | 0.444 | 2.255 | 0.033 |
| | | r | 1.568 | 0.638 | 0.019 |
| | SN | l | 1.104 | 0.905 | 0.022 |
| | | r | 1.537 | 0.651 | 0.019 |
| | VTA | l | 0.254 | 3.936 | 0.038 |
| *Gate closing* | | r | 0.404 | 2.478 | 0.034 |

*Table 4 continued on next page*

*Table 4 continued*

| | ROI | hem | $BF_{10}$ | $BF_{01}$ | Error % |
|---|---|---|---|---|---|
| | Tha | l | 1.722 | 0.581 | 0.018 |
| | | r | 1.138 | 0.879 | 0.022 |
| | Str | l | **3.846** | **0.260** | **$5.752 \times 10^{-7}$** |
| | | r | **3.292** | **0.304** | **$7.157 \times 10^{-7}$** |
| | GPe | l | 1.232 | 0.812 | 0.021 |
| | | r | 0.502 | 1.992 | 0.031 |
| | GPi | l | 2.367 | 0.422 | $1.108 \times 10^{-6}$ |
| | | r | 0.264 | 3.791 | 0.038 |
| | STN | l | 1.486 | 0.673 | 0.019 |
| | | r | **8.206** | **0.122** | **$1.685 \times 10^{-7}$** |
| | SN | l | 0.326 | 3.065 | 0.036 |
| | | r | 0.184 | 5.434 | 0.041 |
| | VTA | l | 0.779 | 1.284 | 0.026 |
| *Substitution* | | r | **4.089** | **0.245** | **$5.267 \times 10^{-7}$** |
| | Tha | l | 1.605 | 0.623 | 0.019 |
| | | r | 1.035 | 0.967 | 0.023 |
| | Str | l | 1.943 | 0.515 | $1.415 \times 10^{-6}$ |
| | | r | 2.233 | 0.448 | $1.192 \times 10^{-6}$ |
| | GPe | l | **90.327** | **0.011** | **$1.536 \times 10^{-8}$** |
| | | r | **3.956** | **0.253** | **$5.523 \times 10^{-7}$** |
| | GPi | l | **4.451** | **0.225** | **$4.633 \times 10^{-7}$** |
| | | r | 0.199 | 5.024 | 0.040 |
| | STN | l | 2.862 | 0.349 | $8.649 \times 10^{-7}$ |
| | | r | **14.117** | **0.071** | **$5.350 \times 10^{-8}$** |
| | SN | l | 0.908 | 1.101 | 0.025 |
| | | r | **3.273** | **0.306** | **$7.215 \times 10^{-7}$** |
| | VTA | l | 0.348 | 2.877 | 0.035 |
| *Updating mode* | | r | 2.185 | 0.458 | $1.225 \times 10^{-6}$ |

opening. We observed no increased activation in the striatum during gate opening but primarily activations in FPN regions, suggesting no active striatal involvement in opening the gate to working memory. In fact, ROI-wise GLMs suggest evidence against the involvement of any basal ganglia nucleus in gate opening. This contrasts with the findings of **Nir-Cohen et al., 2020**, and raises questions about the relationship between the gate opening process in the reference back task and the indirect striatal gating mechanism described in the PBWM model (**Frank et al., 2001**; **Hazy et al., 2007**; **O'Reilly and Frank, 2006**) and other neurocomputational theories (**Hazy et al., 2007**; **Jongkees, 2020**). According to these models, a dopaminergic signal in the striatum is required to trigger gating. Although the orthogonal contrasts in the reference-back task are intended to isolate working memory subprocesses inspired by models of working memory, the two gating contrasts do not fully capture the gating mechanism as originally proposed in neurocomputational models (**Frank et al., 2001**; **Hazy et al., 2007**; **O'Reilly and Frank, 2006**).

However, our finding is partially consistent with the dual-state theory's proposal that dopamine directly modulates PFC representations via mesocortical pathways (**Durstewitz and Seamans, 2008**)

**Table 5.** Results from the Bayesian one-sample t-test on the beta values derived from the region-of-interest (ROI) GLMs using the masks from *Pauli et al., 2018*, are reported for each contrast and ROI. Moderate or higher evidence for ROI activity is indicated by bold font. Bayes factors favoring the alternative hypothesis are reported under $BF_{10}$, and Bayes factors favoring the null hypothesis in column $BF_{01}$.

| | ROI | hem | $BF_{10}$ | $BF_{01}$ | Error % |
|---|---|---|---|---|---|
| Gate opening | VTAnc | l | 0.190 | 5.256 | 0.041 |
| | | r | 0.184 | 5.436 | 0.041 |
| | PBP | l | 0.351 | 2.851 | 0.035 |
| | | r | 0.185 | 5.392 | 0.041 |
| | SNc | l | 0.267 | 3.741 | 0.038 |
| | | r | 0.198 | 5.048 | 0.041 |
| Gate closing | VTAnc | l | 0.388 | 2.578 | 0.034 |
| | | r | 0.213 | 4.705 | 0.040 |
| | PBP | l | 0.340 | 2.944 | 0.035 |
| | | r | 0.414 | 2.415 | 0.033 |
| | SNc | l | 3.348 | 0.299 | $6.990 \times 10^{-7}$ |
| | | r | 32.293 | 0.031 | $4.835 \times 10^{-9}$ |
| Substitution | VTAnc | l | 0.184 | 5.429 | 0.041 |
| | | r | 0.459 | 2.178 | 0.032 |
| | PBP | l | 0.774 | 1.292 | 0.026 |
| | | r | 3.913 | 0.256 | $5.609 \times 10^{-7}$ |
| | SNc | l | 0.754 | 1.326 | 0.027 |
| | | r | 13.330 | 0.075 | $6.138 \times 10^{-8}$ |
| Updating mode | VTAnc | l | 0.224 | 4.474 | 0.040 |
| | | r | 0.343 | 2.916 | 0.035 |
| | PBP | l | 0.265 | 3.777 | 0.038 |
| | | r | 1.304 | 0.767 | 0.021 |
| | SNc | l | 3.105 | 0.322 | $7.757 \times 10^{-7}$ |
| | | r | 142.529 | 0.007 | $2.305 \times 10^{-4}$ |

Note: For all tests, the alternative hypothesis specifies that the population mean differs from 0. $BF_{10}$ 1–3 indicates weak evidence, $BF_{10}$ 3–10 moderate evidence, and $BF_{10}$ >10 strong evidence.

without involving the basal ganglia. Aligning with the dual-state theory (*Durstewitz and Seamans, 2008*), the whole-brain analysis did suggest the involvement of the dopaminergic midbrain in gate opening (*Figure 7*), yet ROI-wise GLM as well as cluster-based ROI analyses could not further corroborate these findings (see appendix). In fact, ROI-wise GLMs indicated moderate evidence against midbrain activity during the process of opening the gate. All analyses thus point to no basal ganglia involvement specific to working memory gate opening. Instead, there is a possibility that mesocortical dopaminergic pathways could regulate gating. However, the evidence for the midbrain involvement remains mixed, leaving the matter inconclusive and implying future work should follow up on the exact role of the dopaminergic midbrain in gate opening. Exploring the connectivity of the meso-limbic pathway during gate opening, for example, could provide valuable insights.

Another explanation for the lack of enhanced striatal activity in gate opening challenges the conceptualization of the gating mechanism in the reference-back task, which, as mentioned above, does not accurately map onto the PBWM predictions. The gate opening contrast includes reference

trials that follow comparison trials regardless of the stimulus match condition. This means that actually gating new, relevant information into the PFC is only necessary on 50% of gate opening trials (i.e. on reference/switch/mismatch trials and not reference/switch/match trials; not to be confused with substitution contrast, which takes into account only repeated trials). Furthermore, in light of the PBWM model's proposal that the basal ganglia sit in a gate-closed state by default, gate opening should take place on every single reference trial. However, engaging in striatal gating every time a reference cue is encountered (i.e. on every reference trial) would be inefficient if the cue is not predictive of the subsequent updating, as is the case for the reference-back task. As a result, this conceptualization of gate opening would be expected to destabilize working memory representations, potentially reducing the accuracy of working memory-based decisions. In support of the idea that striatal gating in response to each reference cue would be an uneconomic brain process, we did not find lower accuracy in reference trials when compared to comparison trials (see *Appendix 1—table 4*), suggesting that gate opening does not occur. Since we, however, did find accuracy costs related to gate opening trials, our findings imply that the costs associated with gate opening might be driven by the trial type/mode switch triggering selective attention rather than reflecting processing costs associated with switching the working memory gate state. That gating costs observed in the reference-back task reflect expenses that are not necessarily associated with mechanisms for switching the gate state is further corroborated by the biology of the basal ganglia, which sit in a default closed-gate (No-Go) state. However, suppose a striatal gating process were to occur in response to every reference trial to facilitate working memory updating. In that case, there should also be striatal/basal ganglia activation in trials where the gate is opened independent of a switch in the gate state. This situation is partially represented in the updating mode, which contrasts repeated reference over repeated comparison trials. If the striatum is only involved when the gate switches from closed to open, no striatal activation should exist in an updating mode.

Intriguingly, we found that the updating mode was associated with strong evidence for basal ganglia engagement, along with strong evidence for the right SNc and right VTA – supported by the right PBP – activity. These findings suggest that repeated reference trials, hence repeated updating cues, engage the basal ganglia – which, in fact, was originally postulated for gate opening by the PBWM model – and that such a sustained updating mode, rather than switching between trial types, causes this activation. Hence, there is a striatal mechanism that is engaged when being prepared for updating is (repeatedly) required. However, this finding is inconsistent with *Nir-Cohen et al., 2020*, who found no evidence for any subcortical involvement in the updating mode contrast. In fact, they found activation contradictory to our results in both the gate opening and updating mode contrasts, rendering future work on the subcortical engagement in these working memory processes essential to understanding the exact neural mechanisms involved. Alternatively, it is also possible that the basal ganglia involvement in the updating mode reflects attentional processes (*Cools et al., 2004*; *Leber et al., 2010*; *van Schouwenburg et al., 2014*) triggered by successive reference trials.

Taken together, our data aligns with the PBWM model, emphasizing the central role of the basal ganglia in working memory processes overall. However, we could not find any evidence for striatal involvement during gate opening, in contrast to previous work (*Nir-Cohen et al., 2020*). Instead, our observations revealed the involvement of FPN regions and accuracy costs during gate opening. These findings suggest that the observed gate opening costs may be better characterized as a selective attention process that does not require striatal selective gating mechanisms.

Moreover, despite the lack of striatal involvement during gate opening, our findings do not rule out the possibility that the PBWM model's predictions about striatal gating in working memory are correct, given the misalignment between the gate opening contrast and the PBWM's proposal regarding striatal gating. It remains unclear whether the absence of striatal activation during gate opening trials is specific to low-demand tasks, like the reference-back task, which does not require as much gating compared to high working memory-demand tasks involving preparation for updating. Or whether the gate opening contrast does not sufficiently capture the PBWM proposed gating mechanism. Further investigation is needed to determine whether (dopamine-driven) striatal gating occurs in high-demand working memory tasks, where the gating process plays a more critical role.

Additionally, our results suggest a more specific role for the basal ganglia in a ready-to-update mode rather than controlling the gate to working memory, as defined in the reference-back task's gate opening contrast. The contrasting patterns of activation observed in the basal ganglia, particularly the

striatum, between the gate opening and updating mode in our dataset compared to the findings of *Nir-Cohen et al., 2020*, highlight an interesting discrepancy and emphasize the importance of additional research to better understand these variations.

## Dopaminergic involvement in working memory substitution

The second aspect of the twofold revelation, indicating a revised role for the basal ganglia in working memory, is the observation of basal ganglia activation in combination with midbrain activation, specifically during substitution trials. Our findings suggest that the striatum – along with the rest of the basal ganglia and the thalamus – is more involved in the actual process of replacing and updating working memory representations ('substitution') than controlling the more general sustained ready-to-update state ('updating mode'). This observation is supported by the enhanced subcortical activity we found in substitution trials evident from whole-brain analyses and ROI-wise GLMs (as well as cluster-based ROI analysis, see Appendix 1). *Nir-Cohen et al., 2023*, further affirm our observation through their modified version of the reference-back paradigm, which includes task switching, demonstrating that the gate opens only when truly necessary (i.e. when updating of the task that is held in working memory is required) and is associated with increased activity in the basal ganglia, thalamus, and midbrain.

We found enhanced neural activation in the dopaminergic midbrain, basal ganglia, thalamus, and PFC when substituting old with new information in working memory. In greater detail, the whole-brain analysis showed large cortical and subcortical activation clusters. Further confirmation from ROI analyses indicated the involvement of the basal ganglia-thalamo-cortical loop in substituting information in working memory. Precisely, ROI-wise GLMs suggested evidence for increased activity in the striatum, right STN, and right VTA, along with activity in the right SNc and right PBP. Furthermore, the cluster-based ROI analysis revealed the involvement of the dlPFC and mPFC, thalamus, and basal ganglia, as well as VTA and SN, both bilaterally notably (see *Appendix 1—table 5*).

It is important to note the differences between the ROI-wise GLMs and the cluster-based ROI analysis (see appendix). The cluster-based ROI analysis examines peak activation in a cluster-based approach, just like whole-brain GLMs, but limited to the ROI. Meanwhile, the ROI-wise GLMs consider the mean signal change over all voxels within a mask. Hence, it is possible that the observed signal change found for both left and right midbrain nuclei in cluster-based ROI analysis for substitution was canceled out in the ROI-wise GLMs, resulting in activation observed only in the right VTA, PBP, and SNc during substitution, but in none of the individually delineated SN mask from MASSP (*Bazin et al., 2020*), which also covers the region associated with the non-dopaminergic SNr. Future studies can further infer dopamine involvement during substitution by tracking the dopaminergic dynamics in binding and release at receptor sites quantified with positron emission tomography (PET) (*Bäckman et al., 2017*). Nonetheless, all three analyses imply engagement of the basal ganglia-thalamo-cortical loop, along with putative dopaminergic neurons of the midbrain, and thus align with our hypothesis that the neural signatures of working memory updating subprocess substitution resemble updating cortical value representations through a (dopaminergic) RPE also via the basal ganglia (*Montague et al., 1996*; *Schultz et al., 1997*; *Schultz, 2013*), and thus the value-updating network associated with reinforcement learning. To what extent these networks overlap precisely represents an exciting avenue for future research employing methods such as joint modeling of functional and behavioral data across tasks (*Palestro et al., 2018*; *Stevenson et al., 2024*).

Finally, our findings indicate a role for dopaminergic brain regions in the substitution of working memory rather than gating, even though it is unclear to what extent working memory gating, as postulated in the PBWM, can be measured accurately by using the reference-back task (see previous section 'No support for striatal gate opening'). Yet, our findings hint toward phasic dopamine representing a signal to alter (in this framework, substitute) representations held in working memory. This suggests that phasic dopamine release signals updating cortical representations independent of the cognitive domain.

Taken together, our results provide convincing evidence for dopaminergic processing when substituting new information into working memory. Moreover, our results suggest shared mechanisms between working memory updating and value-based (reinforcement) learning, both of which update cortical representations in an adaptive manner. Additionally, these findings refine the PBWM model's

gating mechanism and suggest that phasic dopamine release, in fact, may signal the updating of cortical representations.

## Subcortical involvement in gate closing

Based on the whole-brain and cluster-based ROI analyses, we found that the (left) PPC was the primary region involved in gate closing, without involvement from the basal ganglia. This finding is in line with *Nir-Cohen et al., 2020*, and the predictions of the PBWM model (*Frank et al., 2001*; *Hazy et al., 2007*; *O'Reilly and Frank, 2006*). However, it stands in contrast to *Nir-Cohen et al., 2023*, who found that the striatum was involved during distractor conditions that required active gate closing (by filtering out conflicting task cues). Intriguingly, our ROI-wise GLM analyses did reveal additional yet weak support for (right-lateralized) striatal activity during gate closing. Furthermore, our results provide weak evidence for lateralized right STN and SN activation and particularly strong evidence for the right GPe and right SNc activation during the process of gate closing. Across all fMRI analyses, we observed little activation of the PFC and large activation clusters in the PPC. However, given the evidence against thalamic activations, it seems unlikely that the entire basal ganglia-thalamo-cortical loop is engaged during gate closing. Thus, the ROI-wise GLM-based evidence implicating parts of the right basal ganglia in gate closing may hint toward a different functional basal ganglia loop. This is supported by evidence against the activity of the basal ganglia output nucleus GPi and the thalamus in this contrast, suggesting no engagement of the two structures crucial for the signaling between the cortex and basal ganglia.

It is possible that the neural signature of gate closing represents the suppression of inappropriate actions (i.e. working memory updating). Interestingly, the dominance of the right hemisphere in the subcortex observed in the ROI-wise GLMs contrasts with the predominantly left hemisphere findings from the whole-brain analysis. This difference in hemispheric dominance may suggest distinct lateralized functional roles in gate-closing trials. On the one hand, inhibition tasks support a right-lateralized inhibitory network involving the subcortical regions striatum, GPe, and SN (*Maizey et al., 2020*; *Isherwood et al., 2023*). On the other hand, *Maizey et al., 2020*, observed minimal BOLD changes in the expected right cortical area, the inferior frontal gyrus, during active ignorance of updating, potentially explaining the relative prominence of left hemispheric cortical activations in our findings. This pattern is reminiscent of gate-closing trials, possibly indicating an active suppression of updating working memory. In fact, a subset of GPe neurons – which exhibits the strongest evidence of activity in gate closing compared to any other subcortical ROI – projects to the striatum, where they powerfully suppress neurons of the direct and indirect pathway (*Bevan, 2021*; *Glajch et al., 2016*; *Mallet et al., 2012*; *Mallet et al., 2016*). Together, this may indicate an inhibition process within the basal ganglia in our data – supported by the lack of behavioral costs during gate closing, in fact, increased accuracy observed akin to *Boag et al., 2021*, and *Konjusha et al., 2023* – ensuring no erroneous updating signal is transmitted through the basal ganglia-thalamo-cortical loop.

Hence, our findings suggest that the basal ganglia circuit is not engaged in an active gate-closing process due to the lack of evidence involving essential parts of the basal ganglia-thalamo-cortical loop, but instead supports preventing updating related signaling in the subcortex during trials requiring the protection of working memory content. As a result, findings imply the working memory gate is closed by default and that substituting working memory requires active signaling through the otherwise inhibited basal ganglia.

Finally, no evidence for activity in the VTA was found in gate closing, proving no support for gate closing via the mesocortical pathway, as suggested by *Nir-Cohen et al., 2020*, and *Durstewitz and Seamans, 2008*. However, ROI-wise GLMs indicate a potential role of the nigrostriatal pathway originating in the SNc during gate closing, which is a surprising finding. It might represent an erroneous, premature updating signal that engages the subsequent GPe-driven inhibition; however, this remains entirely speculative.

Taken together, our analyses provide evidence against a striatal gate-closing process yet implicate the involvement of other subcortical nuclei (right GPe and the right SNc) during gate closing, which might represent a process to inhibit false updating-related signaling from being transmitted to the cortex. Furthermore, it supports the idea that the working memory gate is closed by default due to the basal ganglia sitting in a No-Go-state, and updating working memory requires actively transmitting a Go-signal through the basal ganglia.

# Functional subdivisions of the dopaminergic midbrain

Gate closing, substitution, and updating mode are differently associated with the activity of the individually parcellated midbrain ROIs VTA and SN. Yet, it remains unclear whether nigrostriatal, mesolimbic, or mesocortical cell populations were involved in the individual contrasts due to limitations in identifying the associated cell assemblies in the individual MRI scan. In the following, dopaminergic-midbrain-associated results across all ROI analyses are discussed in detail in order to potentially shed light on the functional spatial organization of the dopaminergic midbrain region.

For gate closing, in contrast to only very weak evidence from the individual masks by MASSP (*Bazin et al., 2020*), the use of the masks by *Pauli et al., 2018*, revealed bilateral involvement of the SNc in gate closing – particularly very strong evidence for the right SNc – implicating a role for the dopamine neurons of the SN in closing the gate to working memory. The right SNc was also engaged during substitution, in addition to the right PBP, as suggested by ROI-wise GLMs. Interestingly, in contrast to these ROI-wise GLM results for substitution, the cluster-based ROI analysis for substitution using the probabilistic VTA and SN masks from MASSP (*Bazin et al., 2020*) showed broad bilateral activation for both, while analyses using *Pauli et al., 2018* masks implicated only the SNc, bilaterally (*Appendix 1—table 2*). Nonetheless, all three ROI analyses support the idea of dopaminergic modulation of working memory substitution. Overall, the cluster-based ROI analysis revealed activated clusters in the midbrain limited to the substitution contrast, with no activated midbrain clusters in updating mode. However, ROI-wise GLMs using the probabilistic atlas from *Pauli et al., 2018*, revealed strong evidence for the right SNc in updating mode, accompanied by evidence for the left SNc and right PBP. These findings suggest a pronounced involvement of a dopaminergic mechanism in this particular working memory updating mode. On the contrary, no evidence was observed for the VTA nucleus mask by *Pauli et al., 2018*, in any working memory subprocess, which may suggest that this mask does not cover neural populations in the VTA related to working memory updating.

These findings suggest initial evidence for nigrostriatal involvement in gate closing and dopaminergic influence in the control of substituting new information into working memory and a ready-to-update mode. However, the ROI-wise GLMs did not support the bilateral engagement of the SNc in substitution indicated by the cluster-based ROI results and instead additionally implicated the right PBP nucleus. On the one hand, this highlights the differences in the two ROI analysis approaches, using unsmoothed and smoothed data, respectively, as discussed earlier. On the other hand, it challenges the regional specificity suggested if only cluster-based ROI results were considered: Given the directly adjacent location of the two nuclei, it is unclear whether exclusively nigrostriatal neurons are involved during working memory substitution. This ambiguity highlights the necessity of delineating PBP and SNc masks, preferably based on the A10 and A9 cell groups, respectively, in the individual. This is crucial for accurately distinguishing between contributions from the mesolimbic and nigrostriatal pathways (*Haber, 2003*; *Haber and Knutson, 2010*), especially on an individual basis.

Nevertheless, the strong evidence from various ROI analyses for SNc activity in gate closing, substitution, and updating mode clearly points toward the involvement of the nigrostriatal pathway and, thus, the basal ganglia-thalamo-cortical loop associated with motor function. Along these lines, there lies great potential in investigating the individual striatal subdivision to shed light on the contributions of individual dopaminergic pathways in working memory updating processes. The results from both whole-brain GLM and cluster-based ROI analyses indicate that the putamen and caudate are active during substitution and updating mode (please refer to *Appendix 1—table 1*). However, it remains unclear how helpful individual parcellations-based ROI-wise GLMs would be in answering this question if they become available. After all, findings implicate the dorsal striatum, which comprises both the dorsal regions of the putamen and the caudate, which is classically associated with the nigrostriatal pathway (*Haber, 2003*; *Haber and Knutson, 2010*).

Furthermore, the findings associated with working memory substitution suggest meaningful differences between the VTA masks from the two ROI sources. Despite the equal amount of evidence for the VTA and PBP in substitution and updating mode, distinct functional differences associated with the underlying neural populations have become apparent. Specifically, the MASSP VTA mapping includes a large number of voxels that appear to be involved in working memory substitution, while *Pauli et al., 2018* VTA nucleus mask did not elicit activation. Apparently, it does not functionally overlap with the MASSP VTA voxels involved in working memory substitution. The latter is further supported by the finding implicating the right PBP in substitution, reinforcing that the substitution-related activity

is located in the dorsolateral VTA, where the PBP nucleus is situated (*Trutti et al., 2019*). This again underscores the demand for enhanced delineation of the VTA, particularly in distinguishing the dorso-lateral VTA functionally from the ventromedial site.

To summarize, follow-up analyses using different masks associated with dopaminergic cell populations of the midbrain revealed interesting insights. There was strong evidence for the role of the SNc in the process of substitution, supported by both ROI analysis approaches that complement each other's limitations. Also, during gate closing and updating mode, the SNc showed increased activation, suggesting nigrostriatal pathway engagement. Additionally, there was evidence of the right PBP nucleus activity for both substitution and updating mode. At the same time, no contrast revealed evidence for working memory updating-related activity in the VTA nucleus. These observations suggest that the evidence for activity limited to the right VTA using the MASSP masks (*Bazin et al., 2020*) was driven by activity in a region of the midbrain that is associated with both neighboring nuclei, PBP and SNc. The intricacy in the functional engagement by midbrain nuclei emphasizes the necessity for individually parcellated masks, in particular of the VTA, in future studies that provide more precision both with respect to functional and anatomical subdivisions. This becomes especially apparent when considering the likelihood of neighboring nuclei influencing the observed activity (*de Hollander et al., 2017*).

## Limitations

We replicated many behavioral results from previous reference-back studies (e.g. *Boag et al., 2021*; *Jongkees, 2020*; *Rac-Lubashevsky and Kessler, 2016a*; *Rac-Lubashevsky and Kessler, 2016b*; *Rac-Lubashevsky and Kessler, 2018*). Notably, we observed an accuracy gain in gate-closing trials similar to the findings of *Boag et al., 2021*, and *Konjusha et al., 2023*, but inconsistent with *Nir-Cohen et al., 2020*, indicating that an active switch of closing the gate potentially suppresses erroneous updating signal may be advantageous for working memory maintenance. However, we could not replicate the gate-closing cost for RT that other studies have reported (e.g. *Jongkees, 2020*; *Nir-Cohen et al., 2020*; *Rac-Lubashevsky and Kessler, 2016a*; *Rac-Lubashevsky and Kessler, 2016b*; *Rac-Lubashevsky and Kessler, 2018*). This may have been due to our ultrahigh field scanning protocol, which allowed for fewer trials than are typically collected in reference-back studies and included longer intertrial intervals than previous behavioral studies.

Consequently, the fewer trials compared to other work (*Nir-Cohen et al., 2020*; *Nir-Cohen et al., 2023*) may have lowered the fMRI signal-to-noise ratio, potentially resulting in decreased efficacy in detecting neural markers associated with gate opening in the basal ganglia. Nevertheless, compared to other work, our protocol did unveil a substantially broader range of cortical and subcortical activation during other working memory processes, including the involvement of the right thalamus during gate opening, which is a critical part of the basal ganglia-thalamo-cortical loop. Consequently, it remains unclear if this accounts for the present findings. Future work employing ultrahigh field fMRI using a higher trial-number design may shed light on more signals in the subcortical mechanisms involved in the process of gate opening.

Finally, the ROI-wise GLM analysis does not consider striatal subdivisions as these are not parcellated by MASSP. In light of the relatively large volume of the striatum as a whole, the association of the dorsal and ventral striatum with the nigrostriatal and mesolimbic pathways, respectively, in combination with the different functional profiles of the striatal subdivision (*Haber, 2003*), it would be intriguing to explore the contribution of individual striatal subdivision in working memory updating processes in the future, as discussed in the previous paragraph.

## Conclusions

Our finding of the absence of evidence for striatal activity during the process of gate opening, alongside observed activity in FPN regions, suggests the striatum is not crucially involved in opening the gate to working memory as operationalized in the reference-back task. This finding challenges the concept of a striatal working memory gating mechanism in such low-demand working memory tasks and previous empirical results (*Nir-Cohen et al., 2020*; *Nir-Cohen et al., 2023*). Instead, our data revealed basal ganglia engagement associated with the ready-to-update working memory mode ('updating mode'), suggesting a more specific role for the basal ganglia in a sustained ready-to-update mode rather than controlling the gate state to working memory. Moreover, in line with previous work (*Nir-Cohen et al.,*

*2020*), our findings implicate that the basal ganglia-thalamo-cortical loop is not engaged in gate closing, as evidence for basal ganglia output is lacking, despite observed basal ganglia activation. Instead, our ROI analyses, together with behavioral results, indicate the suppression of updating-related signaling in the subcortex during gate closing. Furthermore, our data indicate the involvement of dopamine-producing midbrain nuclei during the process of working memory gate closing, notably only the SNc, and during a general ready-to-update mode, both the SNc and the PBP of the VTA.

In addition, we found evidence suggesting that substituting new information into working memory is driven by dopaminergic midbrain nuclei – notably from all three; SNc, VTA, and PBP – via the basal ganglia and involving the basal ganglia-thalamo-cortical loop. This finding is consistent with neural signatures of value updating triggered by dopaminergic signals via RPE in reinforcement learning, potentially pointing to a common network underlying the updating of cortical representations which could be further quantified with joint modeling in future studies (*Stevenson et al., 2024*). Additionally, connectivity and PET imaging could provide detailed insights to identify a specific dopaminergic pathway, enabling the quantification of the involvement of the mesolimbic and nigrostriatal pathways.

This study furthers understanding of the neural mechanisms underlying working memory updating subprocesses in the human subcortex, providing additional insights into the role of the dopaminergic midbrain. Moreover, it contributes to the development of methodologies for exploring the neurobiological basis of cognitive flexibility and adaptive behavior.

## Methods
### Procedure
37 participants (20 female; mean age 26.65±5.72 years; age range 19–39 years) took part in the study, which was approved by the Ethics Review Board of the Faculty of Social and Behavioral Sciences at the University of Amsterdam, the Netherlands (reference: 2021-BC-13146), and the Regional Committees for Medical and Health Research Ethics, Norway (reference: 116630). All participants provided written informed consent and completed MRI screening forms to ensure they were eligible for scanning. The recruitment was conducted at the Norwegian University of Science and Technology. The participants had a corrected-to-normal vision and no history of epilepsy or overt clinical neuropsychiatric disease. Two participants' data were excluded because they had more than 30% nonresponses or less than 70% accuracy on the reference-back task, suggesting that these participants were either disengaged or misunderstood the task instructions.

### Reference-back task
To disentangle the various working memory updating subprocesses, participants completed the reference-back task (*Rac-Lubashevsky and Kessler, 2016a*). The task required participants to compare a presented stimulus – a capital letter, akin to *Rac-Lubashevsky and Kessler, 2016a* – to a reference stimulus held in working memory (i.e. the referent; *Figure 1*). The color of the stimulus frame indicated whether to open the gate to working memory facilitating potential updating of the referent with the presented stimulus (on red frame/reference trials) or to keep the gate closed to maintain the existing referent (on blue frame/comparison trials). When the presented stimulus matched the referent, participants were instructed to respond 'same' (by pressing the right key). When the presented stimulus did not match the referent, participants were instructed to respond 'different' (by pressing the left key). In other words, reference trials required updating working memory because the current stimulus served as a referent for subsequent trials. Here, the actual act of updating, i.e., the process of replacing old information with new, is represented by substitution. By contrast, comparison trials did not require updating working memory. Working memory gate opening was needed in reference trials that followed comparison trials, and working memory gate closing was required in comparison trials that followed reference trials; hence, both involved a switch in the gate state. Accordingly, the gate state 'switched' or 'repeated' in each trial. This resulted in eight conditions: 2 (trial type: reference vs. comparison)×2 (switch type: repeat vs. switch)×2 (response: same vs. different), which formed the basis of the various reference-back contrasts.

In each block of the task, the trial sequence began with a reference trial that required no response. Following the reference trial, a fixation cross was displayed at the center of the display, followed by a framed letter ('X' or 'O') presented for a duration of 2 s. Participants were instructed to respond

with either 'same' or 'different' during the stimulus presentation phase. Between each pair of stimuli, a fixation cross was presented. The duration of the fixation cross varied randomly and was selected from a pseudo-exponential distribution, with possible durations of 0.75, 1.5, 2.5, or 3 s, in order to decorrelate the design matrix. After the stimulus presentation phase, an intertrial interval was introduced, also in the form of a fixation cross. This intertrial interval had a variable duration ranging from 0.52 to 2.77 s in order to ensure that each trial's total duration was set at four times the TR (repetition time), totaling 5.53 s per trial. This design resulted in a range of total fixation cross durations between 1.27 and 5.77 s for each trial. Within each block, all eight conditions formed by the combination of trial type, switch type, and response were presented 16 times, leading to a total of 256 trials (32 trials for each condition).

In accordance with *Nir-Cohen et al., 2020*, we defined four contrasts based on the eight reference-back conditions (*Table 5*): gate opening, gate closing, substitution, and updating mode. Gate opening was measured by the difference between reference/switch and reference/repeat trials. This is because the process of opening the gate to working memory should only take place in reference/switch trials, as no change in the gate state is required in repeated reference trials. Similarly, gate closing was measured by the difference between comparison/switch and comparison/repeat trials. Substitution takes place in reference/different trials. However, in order to isolate it from any gate-switching effects, the contrast only takes into account 'repeat' trials. Furthermore, in order to set apart the general effects of 'same' and 'different' responses, the difference between 'different' and 'same' responses in comparison trials was used as a baseline. Thus, the cost of substitution is indicated by a larger difference between 'different' and same responses in reference/repeat trials compared to comparison/repeat trials. Additionally, we computed the updating mode contrast (*Nir-Cohen et al., 2020*), operationalized as the difference between reference/repeat and comparison/repeat trials (*Table 5*).

## Behavioral analysis

Statistical tests on all the conditions, including four a priori contrasts of interest, were conducted to examine the effects of different trial types and reference-back measures on mean RT and accuracy.

The first trial from each block, which did not require a response, was excluded, as well as any trials with RT faster than 0.150 s or slower than 3 s, following the exclusion criteria used in *Boag et al., 2021*. For RT analysis, error trials were excluded.

Linear mixed models were employed to assess the statistical significance of trial type (reference/comparison), switch type (switch/repeat), and response (same/different), along with the four contrasts (gate opening, gate closing, substitution, and updating mode) on mean RT and accuracy. General linear mixed models with a Gaussian link function were used for mean RT, and generalized linear mixed models with a probit link function were used for accuracy. Each model included trial type, switch type, and response as fixed effects, along with random intercepts for each participant. We used a significance criterion of alpha equivalent to 0.05.

## MRI data acquisition

Each participant was scanned four times using a 7T Siemens MAGNETOM TERRA scanner (gradient strength = 80 mT/m at 200 T/m/s) equipped with a 32-channel Nova Medical's single channel transmit 32-channel receive head coil. Anatomical scans were collected in the first session, and the remaining sessions involved different functional scans. The data discussed in this article were collected in one of the functional sessions that included four functional runs in which the participants completed two different tasks. One task was the reference-back (*Figure 1*), which consisted of two runs of 129 trials each. The other task was a reversal-learning task and is not analyzed in the present article. The order of the two tasks was randomized between subjects.

The anatomical session involved acquiring a multi-echo gradient-recalled echo scan (GRE) and an MP2RAGE scan. The GRE scan parameters were as follows: TR = 31.0 ms, TE1=2.51 ms, TE2=7.22 ms, TE3=14.44 ms, TE4=23.23 ms, FA = 12°, and FOV = 240 × 240×168 mm$^3$. The MP2RAGE scan parameters were TR = 4300 ms, TE = 1.99 ms, inversions TI1=840 ms, TI2=3270 ms, flip angle 1=5°, flip angle 2=6°, FOV = 240 × 240 × 168 mm$^3$, and bandwidth (BW)=250 Hz/Px (*Marques et al., 2010*).

The experimental session comprised four functional echo-planar imaging (EPI) runs with four EPI volumes acquired with opposite phase encoding directions for susceptibility distortion correction. The functional data were collected using a single-echo 2D-EPI BOLD sequence with the following

parameters: TR = 1380 ms, TE = 14 ms, MB = 2, GRAPPA = 3, voxel size = 1.5 mm isotropic, partial Fourier = 6/8, flip angle = 60°, MS mode = interleaved, FOV = 192 × 192 × 128 mm³, matrix size = 128 × 128, BW = 1446 Hz/Px, slices = 82, phase encoding direction = A >> P, and echo spacing = 0.8 ms. Each of the two tasks of the functional session had a total of two runs, with each run lasting 13 min and 45 s, resulting in a total of four runs and 55 min. To help co-register the functional scans to the high-resolution data from the anatomical session, another anatomical MP2RAGE scan (1 mm) was collected at the end of the functional session.

Physiological data, including heart rate and respiration, were recorded for all participants to assess the impact of physiological noise on the fMRI data.

## MRI data preprocessing

The imaging data was preprocessed using the neuroimaging preprocessing tool *fMRIPrep 20.2.0* (*Esteban et al., 2018*; *Esteban et al., 2019*), which is based on *Nipype 1.7.0* (*Gorgolewski et al., 2011*; *Gorgolewski et al., 2018*; RRID:SCR_002502). The anatomical data preprocessing involved multiple steps such as intensity and nonuniformity correction (using *N4BiasFieldCorrection*, *Tustison et al., 2010*), skull-stripping (using Nipype's *antsBrainExtraction.sh*), and tissue segmentation (using *FSL*'s fast, *Zhang et al., 2001*) of the T1-weighted images. The brain-extracted T1-weighted scans were normalized by means of volume-based spatial nonlinear registration to standard space *ICBM 152 Nonlinear Asymmetricl template version 2009c (MNI152NLin2009cAsym; Fonov et al., 2009*, RRID:SCR_008796) using *antsRegistration* (*ANTs 2.3.3*). For more information on anatomical data preprocessing, see *Miletić, 2023*.

The following preprocessing was performed for each of the two functional (BOLD) runs per task per participant. A reference volume and its skull-stripped version were generated by aligning and averaging one single-band reference (SBRefs). A B0-nonuniformity map (or fieldmap) was estimated based on two EPI references with opposing phase-encoding directions, with *3dQwarp* (*Cox and Hyde, 1997*; AFNI 20160207). Based on the estimated susceptibility distortion, a corrected EPI reference was calculated for a more accurate co-registration with the anatomical reference. The BOLD reference was then co-registered to the T1w reference using *bbregister* (FreeSurfer 6.0.1), which implements boundary-based registration (*Greve and Fischl, 2009*). Co-registration was configured with six degrees of freedom. Head-motion parameters with respect to the BOLD reference (transformation matrices and six corresponding rotation and translation parameters) were estimated before any spatiotemporal filtering using *mcflirt* (FSL 5.0.9, *Jenkinson et al., 2002*). BOLD runs were slice-time corrected using *3dTshift* from AFNI 20160207 (*Cox and Hyde, 1997*; RRID:SCR_005927). A reference volume and its skull-stripped version were generated using a custom methodology of *fMRIPrep*. The BOLD time-series (including slice-timing correction when applied) were resampled onto their original, native space by applying a single, composite transform to correct for head-motion and susceptibility distortions. These resampled BOLD time-series will be referred to as preprocessed BOLD in original space or just preprocessed BOLD. Several confounding time-series were calculated based on the preprocessed BOLD: framewise displacement (FD), 'DVARS' (the spatial standard deviation of difference images), and three region-wise global signals. FD was computed using two formulations following Power (absolute sum of relative motions, *Power et al., 2014*) and Jenkinson (relative root mean square displacement between affines, *Jenkinson et al., 2002*). FD and DVARS were calculated for each functional run, both using their implementations in *Nipype* (following the definitions by *Power et al., 2014*). The three global signals were extracted within the CSF, the WM, and the whole-brain masks. Additionally, a set of physiological regressors was extracted to allow for component-based noise correction (*Behzadi et al., 2007*). Principal components were estimated after high-pass filtering of the preprocessed BOLD time-series (using a discrete cosine filter with 128 s cut-off) for the two *CompCor* variants: temporal (*tCompCor*) and anatomical (*aCompCor*). *tCompCor* components were then calculated from the top 2% variable voxels within the brain mask. For *aCompCor*, three probabilistic masks (CSF, WM, and combined CSF+WM) were generated in anatomical space. The implementation differs from that of *Behzadi et al., 2007*. Instead of eroding the masks by two pixels on BOLD space, the *aCompCor* masks were subtracted from a mask of pixels that likely contain a volume fraction of GM. This mask is obtained by dilating a GM mask extracted from the FreeSurfer's *aseg* segmentation, and it ensures that components were not extracted from voxels containing a minimal fraction of GM. Finally, these masks were resampled into BOLD space and binarized by

thresholding at 0.99 (as in the original implementation). Components were also calculated separately within the WM and CSF masks. For each *CompCor* decomposition, the k components with the largest singular values were retained, such that the retained components' time-series are sufficient to explain 50% of variance across the nuisance mask (CSF, WM, combined, or temporal). The remaining components were excluded. The head-motion estimates calculated in the correction step were also placed within the corresponding confounds file. The confound time-series derived from head motion estimates and global signals were expanded with the inclusion of temporal derivatives and quadratic terms for each (*Satterthwaite et al., 2013*). Frames that exceeded a threshold of 0.5 mm FD or 1.5 standardized DVARS were annotated as motion outliers. All resamplings can be performed with a single interpolation step by composing all the pertinent transformations (i.e. head-motion transform matrices, susceptibility distortion correction when available, and co-registrations to anatomical and output spaces). Gridded (volumetric) resamplings were performed using *antsApplyTransforms* (ANTs), configured with Lanczos interpolation to minimize the smoothing effects of other kernels (*Lanczos, 1964*). Non-gridded (surface) resamplings were performed using *mri_vol2surf* (FreeSurfer). Many internal operations of fMRIPrep use *Nilearn 0.6.2* (*Abraham et al., 2014*, RRID:SCR_001362), mostly within the functional processing workflow.

## Regions-of-interest

For the selection of ROIs, we selected only subcortical masks for our ROI analyses given our main research aim centered on the subcortex. Second, we included individual subcortical masks derived from the MASSP automated parcellation algorithm (*Bazin et al., 2020*) to increase delineation accuracy on a subject level. Third, bilateral masks for each ROI were included to maximize regional specificity. Fourth, the pallidum was included with individual masks for its external (GPe) and internal (GPi) segments. Fifth, playing an important role in subcortical functioning in general and in basal ganglia processes in particular, the STN mask was also selected for this study. Most importantly, we incorporated masks of the dopaminergic midbrain given our secondary research aim focused on the dopamine sources. This resulted in 7 masks (14 bilaterally). Hence, the MASSP algorithm (*Bazin et al., 2020*) was used to parcellate individual anatomical masks for the thalamus (Tha), striatum (Str), GPe and GPi, STN, SN, and VTA.

In addition, for a post hoc ROI analysis we extended the dopaminergic midbrain masks utilizing the subdivision masks based on the probabilistic atlas from *Pauli et al., 2018*. These masks diverge from the delineations by MASSP (*Bazin et al., 2020*) in both shape and volume (*Trutti et al., 2021*), and were constructed with the aim of delineating two functionally different component nuclei of the VTA. Specifically, the nucleus of the VTA (VTAnc) and PBP nucleus. The VTAnc refers to a small ventromedially located nucleus of the VTA, and the PBP mask represents a dorsolateral component nucleus of the VTA. Both VTAnc and PBP masks provided by *Pauli et al., 2018*, fall into the region defined as VTA by MASSP (*Trutti et al., 2019*). In addition, the authors put forward a subdivided SN, providing separate masks of the SNc and SNr, respectively (*Pauli et al., 2018*). Given that both VTA masks and the SNc are associated with dopaminergic cell populations, ROI analyses were repeated employing these three masks.

Furthermore, a supplementary exploration into the BOLD signal clusters within ROIs was carried out for comparison with *Nir-Cohen et al., 2020*. This analysis utilized the whole-brain statistical parametric maps (SPMs) derived from the whole-brain GLMs and, therefore, did not include individual parcellations but, aligning with *Nir-Cohen et al., 2020*, employed probabilistic masks of the striatal subdivision putamen (Pu) and caudate nucleus (Ca) from *Pauli et al., 2018*, instead of the entire striatum from MASSP. The remaining masks were taken from the probabilistic group-level masks from *Bazin et al., 2020*. See the appendix for the cluster-based ROI analysis elaboration (Appendix A1.1).

All masks were registered to *MNI152NLin2009cAsym* using *antsRegistration* (*ANTs 2.3.3*).

## fMRI statistical analysis

The aim of our 7T fMRI study was to shed light on subcortical – in particular possible dopaminergic – contributions to working memory updating subprocesses in the human brain. To achieve this, we extended *Nir-Cohen et al., 2020*, fMRI data analysis procedure, with a specific emphasis on the subcortical regions. Our foundational objective was to compare the results obtained from a protocol optimized for BOLD sensitivity in the subcortex with the current empirical evidence, which has not

definitively established the involvement of subcortical regions. This included a multistep fMRI analysis. First, a whole-brain analysis was conducted to investigate brain activation by means of SPMs for each contrast on a whole-brain level. Second, ROI-wise GLM analysis with increased regional specificity due to extraction from unsmoothed data was conducted to explore the contribution of several subcortical nuclei in working memory updating. Details about the GLM analysis on both the whole-brain and region-specific levels are listed below. Third, a cluster-based ROI analysis was carried out to explore the presence of significant clusters within each ROI, akin to *Nir-Cohen et al., 2020* (for detailed methodology, see A1.2 in the appendix).

A canonical double gamma hemodynamic response function with temporal derivative was employed as the basis set for all methods of analysis (*Glover, 1999*). The design matrix was constructed to encompass the eight experimental conditions associated with the reference-back resulting from the 2 (trial type: reference vs. comparison)×2 (switch type: repeat vs. switch)×2 (response type: same vs. different) factorial design. Error trials were not included in the fMRI analysis. Before conducting the GLM analysis, the functional data underwent high-pass filtering (*Smith and Brady, 1997*), and subsequently, spatial smoothing was applied using SUSAN (kernel-size full-width half-maximum=4.5 mm). Notably, this spatial smoothing factor deviates from the one employed in the study by Nir-Cohen et al. in 2020, where a larger smoothing factor of 6 mm was used. This variation arises due to differences in image acquisition protocols and the improved image resolution and quality in our study, making such extensive smoothing unsuitable for our dataset (*de Hollander et al., 2015*). In addition to the task-specific regressors, our design matrix incorporated six motion parameters (comprising three translational and three rotational parameters), along with DVARS and FD estimates obtained during preprocessing. An 18-regressor RETROICOR model (*Glover et al., 2000*) was employed to model physiological noise. The model comprised a fourth-order phase Fourier expansion of the heart rate signal, a second-order phase expansion of the respiration signal, and a second-order phase Fourier expansion of the interaction between heart rate and respiration (*Harvey et al., 2008*). Additional regressors were used to account for heart rate variability (HRV; *Chang et al., 2009*) and respiratory volume per time unit (RVT; *Birn et al., 2008*; *Harrison et al., 2021*). The PhysIO toolbox (*Kasper et al., 2017*) implemented in the TAPAS software (*Frässle et al., 2021*) was employed to estimate the physiological regressors. For two participants, either both runs or only the second run of physiological data were not collected due to technical issues. In these cases, we substituted the first 20 components obtained through aCompCor, as described by *Behzadi et al., 2007*. Therefore, a total of 36 regressors were used in the model.

Whole-brain analyses were conducted using the FILM method from FSL FEAT (*Jenkinson et al., 2012*; *Woolrich et al., 2001*). These analyses took into account autocorrelated residuals. For the purpose of combining the GLMs at the run level for each task, fixed effects analyses were employed. Group-level models were subsequently estimated using FLAME1+2 from FSL (*Woolrich et al., 2001*). SPMs were generated to visualize the resulting group-level models. The maps were corrected for the family-wise error rate using the random Gaussian field procedure and a critical value of q<0.05 (*Nichols and Hayasaka, 2003*) and a minimum cluster size ($K_E$) of 10 voxels.

In addition, the ROI GLM analysis was conducted as follows: Mean time-series data were extracted from each subcortical ROI first, using probabilistic masks provided by the individual MASSP parcellation (*Bazin et al., 2020*) and second, using the probabilistic (nonindividual) atlas by *Pauli et al., 2018*. Each voxel's contribution to the mean signal of the region was weighted based on its probability of belonging to that region. Subsequently, the time-series data were transformed into percentage signal change values by dividing each timepoint by the mean signal of the time-series, multiplying the result by 100, and then subtracting 100. These time-series data were extracted from unsmoothed data to ensure regional specificity. The individual runs were concatenated for analysis. We infer exclusively from positive BOLD responses, given the disagreements surrounding negative BOLD responses (*Schridde et al., 2008*; *Wade, 2002*). The ROI GLMs thus encompassed ROI-wise GLM's fit on the mean time-series extracted from the unsmoothed functional data of each voxel. Bayesian one-sample t-tests were computed to investigate the evidence for each contrast based on the mean beta (i.e. signal change) derived from the ROI GLMs. For classification, we utilized Bayesian classification by *Jeffreys and Lindsay, 1939*, where Bayes factors between 1 and 3 indicate weak evidence, Bayes factors between 3 and 10 indicate moderate evidence, and Bayes factors greater than 10 indicate strong evidence.

Lastly, by means of providing comparability with the only other existing fMRI study employing the same experimental paradigm (*Nir-Cohen et al., 2020*), we also ran an additional 'cluster-based ROI analysis' which examined each ROI for clusters of activation based on the whole-brain GLMs after correcting for multiple comparisons across the ROI's voxels. Here, we investigated the involvement of specific brain regions previously associated with working memory, including the FPN and the basal ganglia in each working memory subprocess (for methods and results, see A1 in the appendix).

## Code availability statement

All code used for preprocessing and analyses of the data acquired in this study is available at https://osf.io/jmu6q/.

## Acknowledgements

We would like to express our gratitude to Pål Erik Goa for supporting this study by facilitating data acquisition. We also thank Niek Stevenson, Roel van Dooren, and Bryant Jongkees for their valuable contributions to preceding work and for stimulating discussions on the reference-back paradigm. This work was supported by grants from the European Research Council to BUF (8674750) and BH (ERC-2015-AdG-694722), and a Vici grant from the Netherlands Organization for Scientific Research to BUF (016.Vici.185.052). Funding sources were not involved in study design, data collection, and interpretation, or the decision to submit the work for publication.

## Additional information

### Funding

| Funder | Grant reference number | Author |
|---|---|---|
| European Research Council | 8674750 | Birte U Forstmann |
| European Research Council | ERC-2015-AdG- 694722 | Bernhard Hommel |
| Nederlandse Organisatie voor Wetenschappelijk Onderzoek | 016.Vici.185.052 | Birte U Forstmann |

The funders had no role in study design, data collection and interpretation, or the decision to submit the work for publication.

### Author contributions

Anne C Trutti, Conceptualization, Data curation, Software, Formal analysis, Investigation, Visualization, Methodology, Writing – original draft, Writing – review and editing; Zsuzsika Sjoerds, Conceptualization, Supervision, Methodology, Writing – original draft; Russell J Boag, Formal analysis, Investigation, Methodology, Writing – original draft, Writing – review and editing; Solenn LY Walstra, Formal analysis, Investigation, Writing – original draft, Writing – review and editing; Steven Miletić, Conceptualization, Resources, Data curation, Software, Formal analysis, Investigation, Methodology, Writing – original draft, Project administration; Scott JS Isherwood, Conceptualization, Data curation, Investigation, Project administration; Pierre-Louis Bazin, Resources, Data curation, Software, Formal analysis, Investigation, Methodology, Writing – original draft; Bernhard Hommel, Conceptualization, Supervision, Funding acquisition, Investigation, Writing – original draft, Project administration, Writing – review and editing; Sarah Habli, Data curation, Project administration; Desmond HY Tse, Conceptualization, Data curation, Investigation, Methodology; Asta K Håberg, Conceptualization, Data curation, Investigation, Writing – original draft, Project administration, Writing – review and editing; Birte U Forstmann, Conceptualization, Resources, Data curation, Formal analysis, Supervision, Funding acquisition, Investigation, Methodology, Writing – original draft, Project administration, Writing – review and editing

## Author ORCIDs
Anne C Trutti https://orcid.org/0000-0002-0044-4846
Russell J Boag https://orcid.org/0000-0002-7689-0682
Steven Miletić https://orcid.org/0000-0001-7399-2926
Scott JS Isherwood https://orcid.org/0000-0003-2045-9268
Pierre-Louis Bazin https://orcid.org/0000-0002-0141-5510
Asta K Håberg https://orcid.org/0000-0002-9007-1202
Birte U Forstmann https://orcid.org/0000-0002-1005-1675

## Ethics

The study was approved by the Ethics Review Board of the Faculty of Social and Behavioral Sciences at the University of Amsterdam, the Netherlands (reference: 2021-BC-13146), and the Regional Committees for Medical and Health Research Ethics, Norway (reference: 116630). All participants provided written informed consent and completed MRI screening forms to ensure they were eligible for scanning.

Reviewer #1 (Public review): https://doi.org/10.7554/eLife.97874.3.sa1
Reviewer #2 (Public review): https://doi.org/10.7554/eLife.97874.3.sa2
Author response https://doi.org/10.7554/eLife.97874.3.sa3

# Additional files

## Supplementary files

Transparent reporting form

## Data availability

The analyzed data is part of a larger collection that will be made publicly available in the University of Amsterdam's institutional repository (https://uvaauas.figshare.com) once the full dataset is ready for publication. A subset of the data is already publicly available here: https://doi.org/10.21942/uva.25690692.v1. The code for data preprocessing, analysis, and visualization of results is available at https://osf.io/jmu6q/.

The following dataset was generated:

| Author(s) | Year | Dataset title | Dataset URL | Database and Identifier |
|---|---|---|---|---|
| Miletić S, Isherwood SSJ, Trutti AC, Bazin PL, Habli S, DHY Tse, Bazin PL, Håberg AK, Forstmann B | 2025 | WM-updating_subcortex | https://osf.io/jmu6q/ | Open Science Framework, jmu6q |

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

## Appendix 1

### A1. Cluster-based ROI analyses

#### A1.1. Regions-of-interest

Consistent with *Nir-Cohen et al., 2020*, the FPN masks were defined based on cortical maps of BA, yet they were investigated individually in this study. Cortical region parcellations were obtained from the BA segmentations by *Pijnenburg et al., 2021*. Accordingly, the BA 8, BA 9, and BA 46 masks represent the dlPFC, the BA 24 and BA 32 masks represent the medial PFC, and the BA 7 and BA 40 masks represent the PPC. Moreover, we included individual subcortical masks derived from the MASSP automated parcellation algorithm (*Bazin et al., 2020*) to increase delineation accuracy on a subject level, specifically the thalamus, external and internal segments of the pallidum (GPe and GPi, respectively), STN, SN, and VTA. To ensure that the cluster-based ROI analyses were comparable to *Nir-Cohen et al., 2020*, the striatum parcellation was not taken from the MASSP atlas, instead we picked the striatal masks of the caudate nucleus (Ca) and putamen (Pu) from the probabilistic atlas of *Pauli et al., 2018*. For a second cluster-based ROI analysis, selected midbrain (VTAnc, PBP, SNc) masks from the atlas by *Pauli et al., 2018*, were selected.

#### A1.2. fMRI statistical analysis

To assess the replicability of *Nir-Cohen et al., 2020*, findings with our data, this study extended their fMRI data analysis procedure. In addition to a whole-brain analysis, a cluster-based ROI analysis was carried out to explore the presence of significant clusters within each ROI, akin to *Nir-Cohen et al., 2020*. Here, the z-maps produced from the whole-brain GLM were utilized to retrieve the z-values for every voxel inside each ROI on a group level. In an initial step, for every probabilistic map outlining the ROIs, a thresholding was applied to select only those voxels with probability greater than 30% of belonging to the respective region. In a second step, the thresholded masks were binarized. Subsequently, for all ROIs, a voxel-based false discovery rate threshold of p<0.05 (*Yekutieli and Benjamini, 1999*) was applied within each ROI in order to investigate the involvement of the FPN and subcortical regions in working memory updating processes. In short, the cluster-based ROI analysis examined each ROI for clusters of activation based on the whole-brain GLMs after correcting for multiple comparisons across the ROI's voxels.

#### A1.3. Results

Our cluster-based ROI analyses indicated that specific BA associated with the FPN and subcortical ROIs differently contribute to each experimental contrast. The key findings are outlined below for each experimental contrast, but for a detailed breakdown of the contribution of each ROI area, please refer to *Figure 1*, *Appendix 1—tables 1 and 2*.

##### Gate opening

The dlPFC was broadly engaged during gate opening trials as suggested by cluster-based ROI analyses, along with only the involvement of the left mPFC. In the subcortex, a cluster of activity was found in the left thalamus (*Appendix 1—table 1*). The opposing results to ROI-wise GLM analysis (*Table 3*) could be explained by the different cluster-based ROI analysis approaches and suggest that the left thalamus showed higher peaks in activation while other thalamic subnuclei perhaps show a negative change in signal, thus canceling out the positive signal change picked up by the cluster-based ROI analyses.

##### Gate closing

Cluster-based ROI analyses indicated small, lateralized engagement of the dlPFC limited to the left BA 46 and broad activation of the PPC. Intriguingly, the cluster-based ROI analyses did not indicate activity in any subcortical ROI (*Appendix 1—table 1*).

##### Substitution

The entire FPN is associated with the process of substitution as suggested by cluster-based ROI analyses (*Appendix 1—table 1*). The dlPFC is bilaterally activated during substitution trials, and also the mPFC is associated with substitution, but only one cluster, the right BA 32. Cluster-based ROI analyses also indicate activity in PPC regions, including a particularly large cluster in the left BA 7. The results from cluster-based ROI analysis confirmed the involvement of subcortical nuclei in the substitution process, yet with varying contributions. Bilateral activity in the thalamus, caudate, and

putamen was observed, along with bilateral activity in the SN and VTA. Specifically, large clusters were found in the left thalamus and right caudate, and bilateral clusters in the SNc mask by *Pauli et al., 2018*, were revealed (*Appendix 1—table 2*).

## Updating mode

Cluster-based ROI analyses suggest involvement of the entire FPN during updating mode. Precisely, there is evidence for bilateral dlPFC and PPC activity in updating mode trials (*Appendix 1—table 1*), and there was also a small cluster in BA 32 of the left mPFC. Furthermore, results indicated basal ganglia activation, in particular relatively large clusters of activation in the bilateral caudate and putamen, was found, along with a small cluster in left GPe in updating mode contrast.

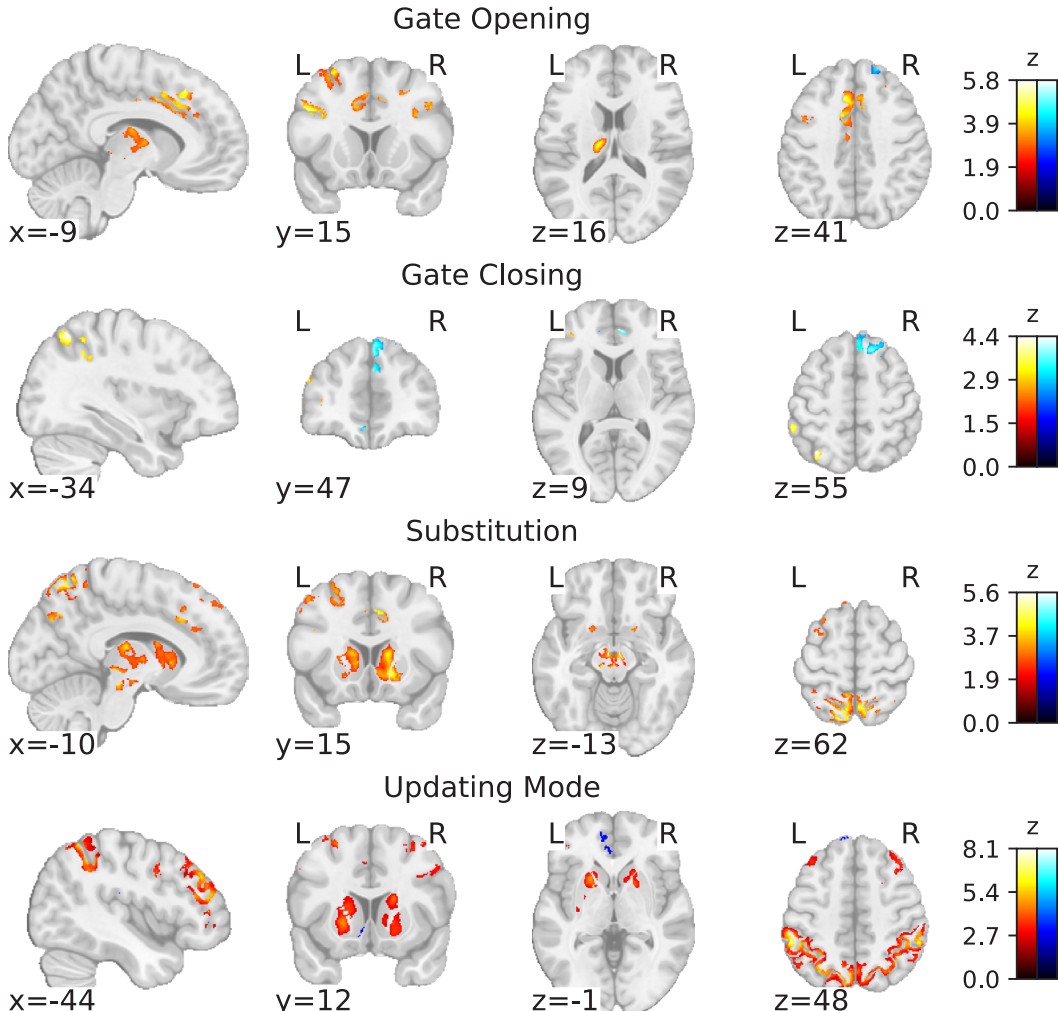

**Appendix 1—figure 1.** Results of the voxel-wise cluster-based region-of-interest (ROI) analysis using ROI-wide false discovery rate (FDR) correction (q<0.05) are illustrated. Each row shows the BOLD signal change within the ROIs for one of the four working memory process contrasts, respectively.

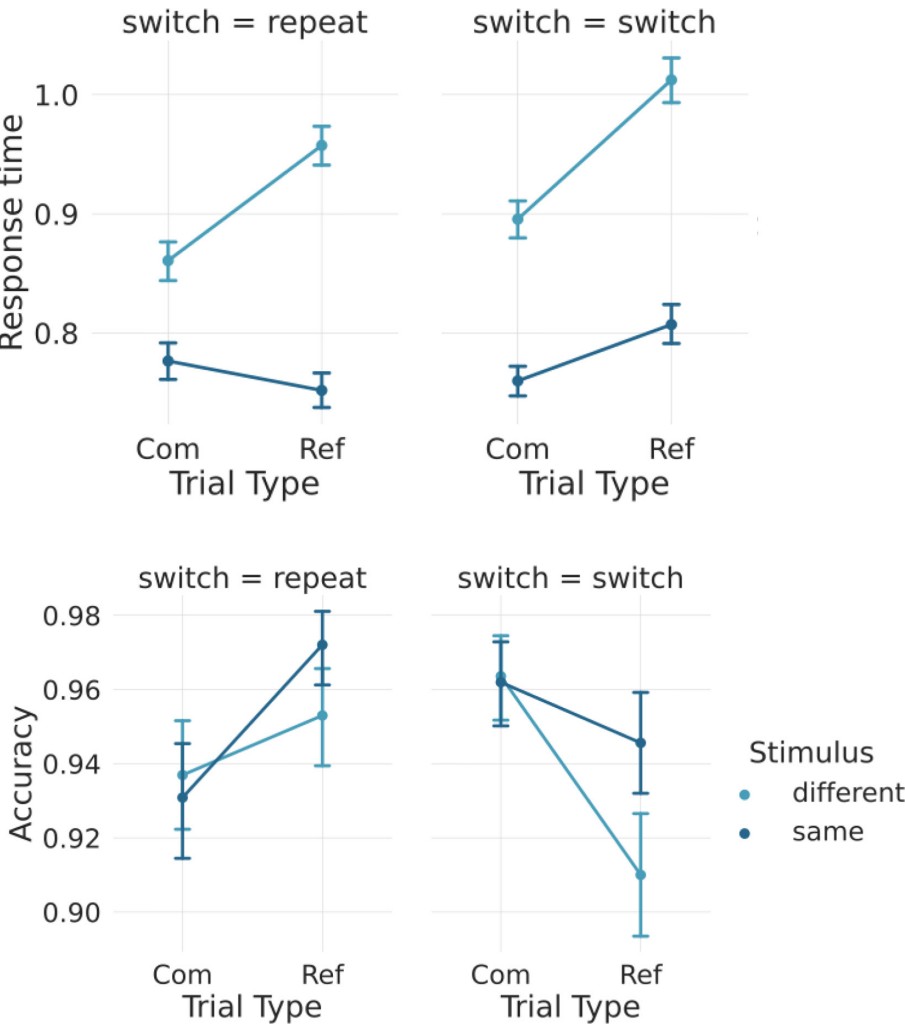

**Appendix 1—figure 2.** The graph outlines the nonsignificant three-way interactions of the factors Trial-type (comparison, reference), Gate-Switch (repeat, switch), and Stimulus/Match (same, different) for both reaction time (RT; top) and accuracy (bottom), respectively. The results indicate that the difference in RT between same and different trials is more pronounced in reference trials compared to comparison trials, suggesting that additional time is required for substitution of the WM item in the reference condition. However, in contrast to previous studies, a larger gate-switching cost, representing the difference between switch and repeat conditions, is only found in reference trials (gate opening) and not in comparison trials (gate closing). The error bars in the figures represent bootstrapped 95% confidence intervals.

The online version of this article includes the following source data for appendix 1—figure 2:

**Appendix 1—figure 2—source data 1.** Behavioral measures from the experimental design.

**Appendix 1—table 1.** List of peak activation in MNI coordinates from the cluster-based region-of-interest (ROI) analysis.

| | ROI | hem | Voxels | MNI | | | |
| | | | | x | y | z | Z |
|---|---|---|---|---|---|---|---|

*Appendix 1—table 1 Continued on next page*

*Appendix 1—table 1 Continued*

|  |  |  |  |  | MNI |  |  |  |
|---|---|---|---|---|---|---|---|---|
| *Gate opening* |  |  |  |  |  |  |  |  |
|  | dlPFC | BA8 | l | 401 | −7.0 | 26.0 | 46.0 | 5.733 |
|  |  | BA8 | r | 205 | 9.0 | 25.0 | 45.0 | 4.667 |
|  |  | BA9 | l | 239 | −46.0 | 15.0 | 32.0 | 5.122 |
|  |  | BA9 | r | 179 | 34.0 | 9.0 | 29.0 | 5.129 |
|  |  | BA46 | l | 94 | −51.0 | 22.0 | 27.0 | 4.746 |
|  | mPFC | BA24 | l | 123 | −9.0 | 4.0 | 40.0 | 4.154 |
|  |  | BA32 | l | 219 | −9.0 | 11.0 | 45.0 | 4.667 |
|  | Subcortex | Tha | l | 401 | −14.0 | −17.0 | 16.0 | 4.613 |
| *Gate closing* |  |  |  |  |  |  |  |  |
|  | dlPFC | BA46 | l | 57 | −50.0 | 42.0 | 9.0 | 4.091 |
|  | PPC | BA7 | l | 306 | −34.0 | −63.0 | 55.0 | 4.392 |
|  |  | BA40 | l | 89 | −45.0 | −47.0 | 39.0 | 4.111 |
| *Substitution* |  |  |  |  |  |  |  |  |
|  | dlPFC | BA8 | l | 258 | −7.0 | 25.0 | 35.0 | 4.704 |
|  |  | BA8 | r | 7 | 4.0 | 15.0 | 42.0 | 4.473 |
|  |  | BA9 | l | 159 | −22.0 | 56.0 | 33.0 | 4.763 |
|  | mPFC | BA32 | r | 119 | 9.0 | 13.0 | 40.0 | 4.533 |
|  | PPC | BA7 | l | 1839 | −14.0 | −70.0 | 62.0 | 5.256 |
|  |  | BA7 | r | 293 | 5.0 | −61.0 | 63.0 | 4.675 |
|  |  | BA40 | l | 335 | −63.0 | −52.0 | 30.0 | 4.822 |
|  | Subcortex | Tha | l | 1151 | −25.0 | −26.0 | −8.0 | 5.129 |
|  |  | Tha | r | 448 | 6.0 | −23.0 | 2.0 | 4.963 |
|  |  | Ca | l | 701 | −14.0 | 11.0 | 10.0 | 4.875 |
|  |  | Ca | r | 1021 | 15.0 | 16.0 | −5.0 | 4.710 |
|  |  | Pu | l | 402 | −20.0 | 5.0 | 0.0 | 5.530 |
|  |  | Pu | r | 296 | 16.0 | 15.0 | −5.0 | 4.791 |
|  |  | GPe | l | 204 | −20.0 | 5.0 | 0.0 | 5.530 |
|  |  | SN | l | 46 | −11.0 | −24.0 | −13.0 | 4.236 |
|  |  | SN | r | 19 | 11.0 | −16.0 | −10.0 | 3.962 |
|  |  | VTA | l | 129 | −2.0 | −22.0 | −19.0 | 4.911 |
|  |  | VTA | r | 145 | 2.0 | −22.0 | −19.0 | 4.349 |
| *Updating mode* |  |  |  |  |  |  |  |  |
|  | dlPFC | BA8 | l | 143 | −26.0 | 5.0 | 60.0 | 5.423 |
|  |  | BA8 | r | 30 | 24.0 | 19.0 | 65.0 | 4.008 |
|  |  | BA9 | l | 832 | −44.0 | 38.0 | 33.0 | 5.816 |
|  |  | BA9 | r | 491 | 32.0 | 38.0 | 26.0 | 4.509 |
|  |  | BA46 | l | 686 | −45.0 | 42.0 | 17.0 | 5.695 |
|  |  | BA46 | r | 525 | 46.0 | 37.0 | 14.0 | 4.238 |

*Appendix 1—table 1 Continued on next page*

*Appendix 1—table 1 Continued*

| | | | | MNI | | | |
|---|---|---|---|---|---|---|---|
| mPFC | BA32 | l | 20 | –4.0 | 10.0 | 45.0 | 3.367 |
| PPC | BA7 | l | 2520 | –7.0 | –77.0 | 52.0 | 6.111 |
| | BA7 | r | 248 | 38.0 | –40.0 | 42.0 | 5.465 |
| | BA40 | l | 2140 | –56.0 | –41.0 | 55.0 | 8.084 |
| | BA40 | r | 1870 | 49.0 | –41.0 | 56.0 | 6.155 |
| Subcortex | Ca | l | 474 | –15.0 | 9.0 | 10.0 | 4.369 |
| | Ca | r | 559 | 17.0 | 13.0 | 16.0 | 4.384 |
| | Pu | l | 445 | –21.0 | 15.0 | –2.0 | 4.983 |
| | Pu | r | 326 | 18.0 | 9.0 | –8.0 | 4.268 |
| | GPe | l | 13 | –17.0 | 9.0 | –1.0 | 3.864 |

**Appendix 1—table 2.** List of peak activation in MNI coordinates from the cluster-based region-of-interest (ROI) analysis using the dopaminergic nuclei' masks from *Pauli et al., 2018*, after false discovery rate (FDR) correction (q<0.05).

The data suggests no activation in the ventral tegmental area (VTA) nucleus and parabrachial pigmented (PBP) nucleus, after FDR correction was applied.

| | | | MNI | | | |
|---|---|---|---|---|---|---|
| ROI | hem | Voxels | x | y | z | Z |
| *Substitution* | | | | | | |
| SNc | l | 16 | –10.0 | –25.0 | –13.0 | 4.489 |
| | r | 56 | 10.0 | –17.0 | –10.0 | 4.178 |

**Appendix 1—table 3.** Significance testing of mean response time (RT) effects using a general linear mixed model with a Gaussian link function.

| Effect | df | F | p |
|---|---|---|---|
| Trial type | 1, 36.20 | 73.520 | <0.001 |
| Switch type | 1, 44.45 | 32.521 | <0.001 |
| Response type | 1, 33.82 | 263.189 | <0.001 |
| Trial type × switch type | 1, 56.82 | 19.802 | <0.001 |
| Trial type × response type | 1, 33.59 | 29.062 | <0.001 |
| Switch type × response type | 1, 166.02 | 6.005 | 0.015 |
| Trial type × switch type × response type | 1, 33.17 | 3.836 | 0.059 |

Note: Type III sum of squares.
Trial type = reference/comparison, match = same/different, switch = repeat/switch.

**Appendix 1—table 4.** Significance testing of accuracy effects using a generalized linear mixed model with a probit link function.

| Effect | df | F | p |
|---|---|---|---|
| Trial type | 1 | 0.167 | 0.683 |
| Switch type | 1 | 5.666 | 0.017 |
| Response type | 1 | 0.306 | 0.580 |
| Trial type × switch type | 1 | 9.996 | 0.002 |

*Appendix 1—table 4 Continued*

| Effect | df | F | p |
|---|---|---|---|
| Trial type × response type | 1 | 42.588 | <0.001 |
| Switch type × response type | 1 | 0.086 | 0.770 |
| Trial type × switch type × response type | 1 | 0.007 | 0.934 |

Note: Type III sum of squares.

Trial type = reference/comparison, match = same/different, switch = repeat/switch.

**Appendix 1—table 5.** Inferential statistics of the response time and accuracy data analysis.

| | | Estimate | SE | df | z | p* |
|---|---|---|---|---|---|---|
| Response time | | | | | | |
| | Gate opening | 0.120 | 0.018 | 34 | 6.547 | <0.001 |
| | Gate closing | 0.014 | 0.015 | 34 | 0.949 | 0.797 |
| | Substitution | 0.125 | 0.020 | 34 | 6.107 | <0.001 |
| | Updating mode | 0.070 | 0.015 | 34 | 4.598 | <0.001 |
| Accuracy | | | | | | |
| | Gate opening | –0.062 | 0.014 | ∞ | –4.463 | <0.001 |
| | Gate closing | 0.049 | 0.013 | ∞ | 3.802 | <0.001 |
| | Substitution | –0.021 | 0.012 | ∞ | –1.792 | 0.073 |
| | Updating mode | 0.048 | 0.013 | ∞ | 3.726 | <0.001 |

*p-Values are adjusted using Holm adjustment.

