## [Editor Report · eLife Assessment]

This **valuable** study uses high-field fMRI to test the hypothesized involvement of subcortical structures, particularly the striatum, in updating working memory. The study overcomes limitations of prior work by applying high-field imaging with a more precise definition of regions of interest in the brain. Thus, the empirical observations are of use to specialists interested in working memory gating or the reference back task specifically. The evidence is generally **solid**, but strong conclusions on dopaminergic contributions must await additional work using molecular imaging or related techniques.

---

## [Referee Report · Reviewer #1 (Public review)]

Summary:

Trutti and colleagues used 7T fMRI to identify brain regions involved in subprocesses of updating the content of working memory. Contrary to past theoretical and empirical claims that the striatum serves a gating function when new information is to be entered into working memory, the relevant contrast during a reference-back task did not reveal significant subcortical activation. Instead, the experiment provided support for a role of subcortical (and cortical) regions in other subprocesses.

Strengths

The use of high-field imaging optimized for subcortical regions in conjunction with the theory-driven experimental design mapped well to the focus on a hypothetical striatal gating mechanism.

Consideration of multiple subprocesses and the transparent way of identifying these, summarized in a table, will make it easy for future studies to replicate and extend the present experiment.

Weaknesses:

The reference-back paradigm seems to only require holding a single letter in working memory (X or O; Fig 1). It remains unclear how such low demand on working memory influences associated fMRI updating responses. It is also not clear whether reference-switch trials with 'same' response truly taxes working-memory updating (and gate opening), as the working-memory content/representation does not need to be updated in this case. These potential design issues, together with the rather low number of experimental trials, raise concerns about the demonstrated absence of evidence for striatal gate opening. Adding an experiment with higher working-memory demand and additional trials could strengthen the evidence for the authors present claim

The authors provide a motivation for their multi-step approach to fMRI analyses. Still, the three subsections of fMRI results (3.2.1; 3.2.2; 3.3.3) for 4 subprocesses each (gate opening, gate closing, substitution, updating mode) made the Results section complex and it was not always easy to understand why some but not other approaches revealed significant effects (as the midbrain in gate opening).

It could be helpful to readers to further revise the Results section and/or more clearly convey the analytic strategy.

The many references to the role of dopamine are interesting, but the discussion of dopaminergic pathways and signals remains speculative and must be confirmed in future studies (e.g., with PET imaging).

Several relevant studies were not cited (e.g., Dahlin et al., 2008, Science; Bäckman et al., 2011, Science).

---

## [Referee Report · Reviewer #2 (Public review)]

Summary:

The study reported by Trutti et al. uses high-field fMRI to test the hypothesized involvement of subcortical structure, particularly striatum, in WM updating. Specifically, participants were scanned while performing the Reference Back task (e.g., Rac-Lubashevsky and Kessler, 2016), which tests constructs like working memory gate opening and closing and substitution. While striatal activation was involved in substitution, it was not observed in gate opening.

While there have been prior fMRI studies of the reference back task (Nir-Cohen et al., 2020), the present study overcomes limitations in prior work, particularly with regard to subcortical structures, by applying high-field imaging with more precise definition of ROIs. And, the fMRI methods are careful and rigorous, overall. Thus, the empirical observations here are useful and will be of interest to specialists interested in working memory gating or the reference back task specifically. I do not have additional concerns about this contribution.

---

## [Author Response]

The following is the authors’ response to the original reviews.

**eLife Assessment**
This useful study uses high-field fMRI to test the hypothesized involvement of subcortical structure, particularly the striatum, in WM updating. It overcomes limitations in prior work by applying high-field imaging with a more precise definition of ROIs. Thus, the empirical observations are of use to specialists interested in working memory gating or the reference back task specifically. However, evidence to support the broader implications, including working memory gating as a construct, is incomplete and limited by the ambiguities in this task and its connection to theory.

We would like to express our gratitude to the editor and the reviewers for their time and effort in providing insightful and valuable comments. We greatly value the critical perspective on the relationship between fMRI contrasts and the PBWM model. We hope to have addressed all the last critical points and changed the manuscript according to the reviewers’ suggestions. Furthermore, we would like to point out that the behavioral results section was edited, as a double-check of the results section revealed some erroneous descriptive statistics.

**Public Reviews:**

**Reviewer #1:**
Summary:Trutti and colleagues used 7T fMRI to identify brain regions involved in subprocesses of updating the content of working memory. Contrary to past theoretical and empirical claims that the striatum serves a gating function when new information is to be entered into working memory, the relevant contrast during a reference-back task did not reveal significant subcortical activation. Instead, the experiment provided support for the role of subcortical (and cortical) regions in other subprocesses.Strengths:The use of high-field imaging optimized for subcortical regions in conjunction with the theory-driven experimental design mapped well to the focus on a hypothetical striatal gating mechanism.Consideration of multiple subprocesses and the transparent way of identifying these, summarized in a table, will make it easy for future studies to replicate and extend the present experiment.Weaknesses:The reference-back paradigm seems to only require holding a single letter in working memory (X or O; Figure 1). It remains unclear how such low demand on working memory influences associated fMRI updating responses. It is also not clear whether reference-switch trials with 'same' response truly tax working-memory updating (and gate opening), as the working-memory content/representation does not need to be updated in this case. These potential design issues, together with the rather low number of experimental trials, raise concerns about the demonstrated absence of evidence for striatal gate opening.

We acknowledge that a limitation of our study is that the task involved relatively low working memory demands. It remains to be clarified whether the same neural mechanisms would be engaged under a higher working memory load, and this is an important consideration for future research.

We also fully agree that it is uncertain whether reference-switch trials requiring a ‘same’ (or ‘match’) response truly engage working memory updating (or gate opening), as the working memory content or representation does not need to be altered in these cases. This concern is addressed in detail in the discussion section titled “No Support for Striatal Gate Opening” (see second paragraph).

Regarding our references to dopamine, we completely agree with the reviewer about the speculative nature of these discussions. In response, we thoroughly reviewed the manuscript and made revisions where necessary to ensure that we consistently emphasize the speculative nature of our commentary on dopamine and dopaminergic pathways.

Finally, we acknowledge the concerns about the design and the relatively low number of trials. However, our fMRI analyses of other reference-back task contrasts did reveal activity in the striatum and other subcortical ROIs. This suggests that our scanning protocol and task design are sufficiently sensitive to detect striatal activity, even with the limited number of trials.

The authors provide a motivation for their multi-step approach to fMRI analyses. Still, the three subsections of fMRI results (3.2.1; 3.2.2; 3.3.3) for 4 subprocesses each (gate opening, gate closing, substitution, updating mode) made the Results section complex and it was not always easy to understand why some but not other approaches revealed significant effects (as the midbrain in gate opening).

We thank the reviewer for this important remark and the opportunity to clarify our approach. We conducted whole-brain general linear models (GLMs) to generate a comprehensive wholebrain map of brain activity for each contrast. However, the whole-brain statistical parametric mappings (SPMs) involve data smoothing, which–while improving signal detection–reduces spatial precision. This is especially problematic in smaller or closely adjacent regions, where spatial blurring can merge distinct activations or make localized signals appear more widespread.

Additionally, the statistical thresholds in whole-brain analyses may detect weak or borderline significant effects, whereas ROI-wise GLMs, which assume uniform behavior across the entire region, may miss the same effects if the signal is weak or inconsistent across the ROI.

Since our primary focus was on the subcortex, we relied more heavily on ROI-wise GLMs, which were limited to subcortical regions. We prioritized findings that were supported by either the ROI-wise GLMs or by both GLM analyses. For instance, the midbrain activations found in our whole-brain analysis but not in the ROI analysis may result from smoothing (where activation from neighboring regions spreads into midbrain voxels) or from functional heterogeneity within the ROI, which can obscure localized activations when averaged in the ROI-wise GLMs. Inferences from each GLM approach, along with their discrepancies, are discussed for each contrast throughout the discussion, with additional details on the clusterbased ROI analysis in the discussion section titled “Dopaminergic involvement in working memory substitution” (see third paragraph).

We acknowledge that the results section may seem complex, and we apologize for any inconvenience this may cause.

**Reviewer #2:**
Summary:The study reported by Trutti et al. uses high-field fMRI to test the hypothesized involvement of subcortical structure, particularly striatum, in WM updating. Specifically, participants were scanned while performing the Reference Back task (e.g., Rac-Lubashevsky and Kessler, 2016), which tests constructs like working memory gate opening and closing and substitution. While striatal activation was involved in substitution, it was not observed in gate opening. This observation is cited as a challenge to cortico-striatal models of WM gating, like PBWM (Frank and O'Reilly, 2005).Strengths:While there have been prior fMRI studies of the reference back task (Nir-Cohen et al., 2020), the present study overcomes limitations in prior work, particularly with regard to subcortical structures, by applying high-field imaging with a more precise definition of ROIs. And, the fMRI methods are careful and rigorous, overall. Thus, the empirical observations here are useful and will be of interest to specialists interested in working memory gating or the reference back task specifically.Weaknesses:I am less persuaded by the more provocative points regarding the challenge it presents to models like PBWM, made in several places by the paper. As detailed below, issues with conceptual clarity of the main constructs and their connection to models, like PBWM, along with some incomplete aspects of the results, make this stronger conclusion less compelling.(1) The relationship of the Nir-Cohen et al. (2020) task analysis of the reference back task, with its contrasts like gate opening and closing, and the predictions of PBWM is far from clear to me for several reasons.First, contrasts like gate opening and gate closing make strong finite state assumptions. As far as I know, this is not an assumption of PBWM, certainly not for gate opening. At a minimum, PBWM is default closed because of the tonic inhibition of cortico-thalamic dynamics by the globus pallidus. Indeed, this was even noted in the discussion of this paper, which seems to acknowledge this discrepancy, but then goes on to conclude that they have challenged the PBWM model anyway.

We thank the reviewer for this remark and agree that the reference-back task contrasts do not perfectly align with the predictions of the PBWM model. In the discussion section "No support for striatal gate opening," we note that our data support the PBWM model by emphasizing the central role of the basal ganglia in working memory processes. However, we acknowledge that it may not have been sufficiently clear in the manuscript that the way the reference-back task is operationalised does not allow for a precise test of the PBWM's gating predictions. To address this, we have revised the manuscript to shift focus away from framing it as a direct challenge to the PBWM model. Below, some edits are highlighted.

‘This contrasts with the findings of Nir-Cohen et al. (2020) and raises questions about the relationship between the gate opening process in the reference back task and the indirect striatal gating mechanism described in the PBWM model (Frank et al., 2001; Hazy et al., 2007; O’Reilly & Frank, 2006) and other neurocomputational theories (Hazy et al., 2007; Jongkees, 2020). According to these models, a dopaminergic signal in the striatum is required to trigger gating. Although the orthogonal contrasts in the referenceback task are intended to isolate working memory subprocesses inspired by models of working memory, the two gating contrasts do not fully capture the gating mechanism as originally proposed in neurocomputational models (Frank et al., 2001; Hazy et al., 2007; O’Reilly & Frank, 2006).’ (line 721-730)

‘Another explanation for the lack of enhanced striatal activity in gate opening challenges the conceptualization of the gating mechanism in the reference-back task, which does not accurately map onto the PBWM predictions.’ (line 746)

‘Moreover, despite the lack of striatal involvement during gate opening, our findings do not rule out the possibility that the PBWM model's predictions about striatal gating in working memory are correct, given the misalignment between the gate opening contrast and the PBWM’s proposal regarding striatal gating. It remains unclear whether the absence of striatal activation during gate opening trials is specific to low-demand tasks, like the reference-back task, which does not require as much gating compared to high working memory-demand tasks involving preparation for updating. Or whether the gate opening contrast does not sufficiently capture the PBWM proposed gating mechanism. Further investigation is needed to determine whether (dopamine-driven) striatal gating occurs in high-demand working memory tasks, where the gating process plays a more critical role.’

Second, as far as I know, PBWM emphasizes go/no-go processes around constructs of input- and output-gating, rather than state shifts between gate opening and closing. While this relationship is less clear in reference back, substituting task-relevant items into working memory does appear to be an example of input gating, as modeled by PBWM. Thus, it is not clear to me why the substitution contrast would not be more of a test of input gating than the gate opening contrast, which requires assumptions that are not clear are required by the model, as noted above.

We fully agree with the reviewer, which is why we proposed that neural mechanisms involving the midbrain and striatum are more likely to be observed in the substitution contrast rather than the gate opening contrast.

Third, PBWM relies on striatal mechanisms to solve the problem of selective gating, inputting, or outputting items in memory while also holding on to others. Selective gating contrasts with global gating, in which everything in memory is gated or nothing. The reference back task is a test of global gating. It is an important distinction because non-striatal mechanisms that can solve global gating, cannot solve selective gating. Indeed, this limitation of non-striatal mechanisms was the rationale for PBWM adding striatum. The connectivity of the striatum with the cortex permits this selectivity. It is not clear that the reference back task tests these selective demands in the first place. That limitation in this task was the rationale behind the recent Rac-Lubashevsky and Frank (2022) paper using the reference back 2 procedure that modifies the original reference back for selective gating.

We thank the reviewer for highlighting this excellent reference. We believe it holds exciting potential for future high-field fMRI studies that explore the neural mechanisms underlying selective gating.

So, if the primary contribution of the paper is to test PBWM, as suggested by the first line of the abstract, then it is not clear that the reference back task in general, or the gate opening contrast in particular, is the best test of these predictions. Other contrasts (substitution), or indeed, tasks (reference back 2) would have been better suited.

We agree with the reviewer that the gate opening contrast may not be the optimal test for the PBWM model predictions. However, previous studies have found evidence of striatal gateopening mechanisms using the reference-back task, which cannot be overlooked. We hypothesized that striatal mechanisms are likely active only when working memory content requires replacement, as seen in the substitution contrast in line with the PBWM model. Additionally, the reference-back 2 task (Rac-Lubashevsky & Frank, 2021) had not yet been published when we began data collection. Exploring this task in future studies, particularly with a 7 T fMRI protocol optimized for subcortical regions, would be an exciting avenue for further investigation.

Finally, in response to the reviewer’s remark, we have revised the abstract to remove the emphasis on challenging the PBWM model.

(2) In general, observations of univariate activity in the striatum have been notoriously variable in the context of WM. Indeed, Chatham et al. (2014) who tested working memory output gating - notably in a direct test of the predictions of PBWM - noted this variability. They too did not observe univariate activation in the striatum associated with selective output gating. Rather they found evidence of increased connectivity between the striatum and cortex during selective output gating. They argued that one account of this difference is that striatal gating dynamics emerge from the balance between the firing of both Go and NoGo cell populations that decide whether to gate or not. It is not always clear how this balance should relate to univariate activation in the striatum. Thus, the present study might also test cortico-striatal connectivity, rather than relying exclusively on univariate activation, in their test of striatal involvement in these WM constructs.

We appreciate the reviewer’s insightful observation regarding the variability of univariate activity in the striatum, particularly in the context of working memory and the challenges noted by Chatham et al. (2014). We agree that striatal gating dynamics likely reflect a balance between Go and NoGo cell populations, which may not always manifest in univariate activation alone. In line with the reviewer’s suggestion, examining cortico-striatal connectivity could provide a more comprehensive understanding of striatal involvement in working memory processes, particularly selective gating.

While our current study focused primarily on univariate activity, we recognize the importance of connectivity-based approaches and plan to incorporate functional connectivity analyses in future studies to further explore these dynamics. Such an approach, especially when combined with ultra-high-field fMRI, may offer valuable insights into the interaction between the striatum and cortex during working memory tasks.

(3) It is concerning that there was no behavioral cost for comparison switch vs. repeat trials. This differs from with prior observations from the reference back (e.g., Nir-Cohen et al., 2020), and in general, is odd given the task switch/cue interpretation component. This failure to observe a basic behavioral effect raises a concern about how participants approached this task and how that might differ from prior reports of the reference back. If they were taking an unusual strategy, it further complicates the interpretation of these results and the implications they hold for theory.

We understand the reviewer’s concern regarding the lack of behavioral response time costs for comparison switch versus repeat trials, which does indeed differ from previous findings in studies such as Nir-Cohen et al. (2020). It is possible that this results from our fMRI task design, such as increased inter-trial intervals compared to behavioral studies. While this is certainly a point of concern, we believe that the neural data still provide valuable insights into the mechanisms underlying working memory gating despite the absence of a clear behavioral effect.

In future studies, we aim to increase the number of trials and more closely align our task design with previous studies to mitigate this issue. We agree that further investigation is necessary to ensure the robustness of these effects and their theoretical implications.

In summary, the present observations are useful, particularly for those interested in the reference back task. For example, they might call into question verbal theories and task analyses of the reference back task that tie constructs like gate-opening to striatal mechanisms. However, given the ambiguities noted above, the broader implications for models like PBWM, or indeed, other models of working memory gating, are less clear.